# A conformal piezoelectric microsystem for demographic-adaptive and calibration-free cuffless blood pressure monitoring

Cunman Liang[1,2,3,6], Zhou Jiang[1,2,4,6], Shirong Qiu[1,6], Lei Zhao[1,2,4], Xinxin Mao[1], Xiao Yang [2], Yuanting Zhang[1,5] & Ni Zhao [1,2] ✉

Frequent recalibration poses a significant challenge to the accuracy and reliability of current cuffless blood pressure (BP) monitoring devices, especially in long-term use. Here, we introduce a conformal and stretchable piezoelectric microsystem (CSPM) combined with a demographic-adaptive BP estimation model, enabling calibration-free, continuous BP tracking with accuracy comparable to cuff-based medical devices. The CSPM integrates an ultrasound transducer array for vessel diameter waveform recording and two pulse waveform (PWF) sensors with cavity-enhanced piezoelectric thin films for precise pulse wave detection, enabling simultaneous measurement of pulse wave velocity (PWV) and vessel diameter in the same localized vascular area. Leveraging real-time PWV and vessel diameter inputs, our BP algorithm employs a demographic feature learning module for broad population applicability and a time decay compensation strategy to address minor slippage and measurement shifts, enabling the acquisition of individual BP values without calibration. This innovation effectively addresses key challenges in population recalibration and multimodal integration, enabling stable long-term continuous BP tracking, achieving mean absolute errors of 5.22 mmHg for systolic BP and 4.57 mmHg for diastolic BP.

Cuffless blood pressure (BP) measurement technologies represent a pivotal advancement in healthcare monitoring, offering a paradigm shift from traditional cuff-based methods[1-3]. These technologies enable non-invasive, continuous BP monitoring by integrating innovative sensors and advanced BP models, thus providing real-time insights into cardiovascular dynamics, enabling early hypertension detection and personalized healthcare[4-7]. Various cuffless BP measurement techniques have been developed, including pressure sensor-based tonometry[8-10], optical methods such as photoplethysmography (PPG)[11-13], electrical methods like impedance plethysmography (IPG)[14,15], and acoustic methods, including ultrasound wall-tracking[16-19] and resonance sonomanometry[20] (see Supplementary Note 1). Despite

these advancements, these methods face significant challenges in practical use. Limited in their capacity to capture a comprehensive set of hemodynamic parameters, they often fall short in achieving consistent BP measurement accuracy, especially over prolonged monitoring periods and across diverse populations. This limitation frequently necessitates recalibration, as BP algorithms struggle to provide accurate readings when applied to the same individual at different time points or across diverse populations, posing a substantial barrier to the practical deployment of cuffless BP measurement technology.

To address these challenges, two primary approaches have been taken: one focusing on advancements in models and algorithms, and

---

[1]Department of Electronic Engineering, The Chinese University of Hong Kong, Hong Kong, China. [2]Hong Kong Centre for Cerebro-cardiovascular Health Engineering (COCHE), Hong Kong, China. [3]Key Laboratory of Mechanism Theory and Equipment Design of Ministry of Education, Tianjin University, Tianjin, China. [4]Department of Biomedical Engineering, City University of Hong Kong, Hong Kong, China. [5]AICARE Bay Lab at Guangdong Medical University, Dong-Guan, China. [6]These authors contributed equally: Cunman Liang, Zhou Jiang, Shirong Qiu. ✉e-mail: nzhao@ee.cuhk.edu.hk

the other on innovations in device design. The development of hemodynamic models and fusion strategies for BP estimation has improved the reliability and adaptability of cuffless monitoring techniques[21–24]. Several BP factors and calibration strategies have been explored to mimic the components of the cardiovascular system and their interactions. These factors and strategies include local vascular resistance[21,22], blood volume perfusion and distribution[23–25], the cardiovascular coupling effects[26], hydrostatic maneuver effects[27], and demographic alignments (e.g., age, sex)[28]. These efforts have expanded upon traditional pulse transit time (PTT)-based models[29–31]. Additionally, data-driven approaches leveraging machine learning and deep learning have integrated multiple hemodynamic parameters, utilizing extensive datasets from diverse physiological signals[32–34]. These techniques enhance model precision by identifying intricate patterns and improving real-time estimation accuracy. Despite these advances, frequent recalibration remains necessary in physiological cuffless BP methods due to oversimplified blood vessel models and limited alignment strategies[35]. While machine learning-based BP estimation algorithms show promise for calibration-free monitoring, their adaptability to untrained subject groups and long-term stability require further investigation[34,36].

On the device front, ultrathin sensors, such as graphene-based sensor, offer improved skin adherence, minimizing motion-induced shifts and slippage that often necessitate recalibration. This advancement significantly enhances measurement stability during dynamic conditions[15]. However, while these sensors address recalibration needs within a single test, they do not eliminate the requirement for recalibration across different testing sessions. Flexible ultrasound sensors offer precise measurements of vascular diameter changes, translating them into accurate BP readings[16,37]. To address alignment challenges, these sensors have been engineered into closely spaced arrays, ensuring consistent measurement precision across repeated tests without the need for recalibration[19,38]. Nevertheless, this method is not sufficient to address variations in vascular conditions across different demographic groups or over extended periods. Devices with multiple sensors, capable of measuring various parameters concurrently, leverage interparameter relationships and self-calibration to potentially minimize calibration requirements. Some are based on the PTT principle or machine learning algorithms for measurements, which still fail to address the issue of repeated calibration at the model level[39–41]. Other multimodal devices integrate PPG, ultrasound sensor or pressure sensor, and involve the use of handheld probes or wristband watches, which necessitate the application of pressure during testing[31,42,43]. This complicates matters by affecting vascular conditions, resulting in inaccuracies in BP modeling. Additionally, pressure application can disrupt other sensor measurements, causing unwanted signal crosstalk and necessitating repeated calibration for accuracy.

Here, we adopted a holistic approach that combines innovations in both model enhancements and device design to address the challenges of calibration and long-term reliability. From this perspective, we developed a conformal and stretchable piezoelectric microsystem (CSPM) capable of capturing multiple hemodynamic parameters, coupled with a demographic-adaptive BP algorithm for calibration-free cuffless BP monitoring. This integrated system enables continuous BP tracking with an accuracy comparable to that of professional medical equipment, even during long-term use. The CSPM incorporates two pulse waveform (PWF) sensors and an ultrasound transducer array, allowing for the simultaneous detection of local pulse wave velocity (PWV) and vessel diameter waveform within the same localized vascular region. The PWF sensors, utilizing piezoelectric thin films for swift responses and featuring a cavity structure to enhance sensitivity, accurately monitor pulse waves on the skin surface. Meanwhile, the ultrasound sensor configured in a 2 × 4 ultrasound transducer array, with 1.6 mm spacing for precise mapping and

a 5 MHz resonant frequency, ensures accurate vessel diameter measurements. The BP algorithm features a dynamic demographic feature learning module to extract cardiovascular characteristics relevant to BP across diverse populations, ensuring broad applicability. Additionally, a time decay compensation strategy addresses structural distortions from minor slippage and localized shifts in the CSPM, ensuring precise continuous BP waveform measurements. Compared to existing cuffless BP devices, our approach overcomes challenges related to population recalibration, multimodal skin integration, and stable long-term tracking. This work demonstrates significant advantages in wearability, continuous monitoring, and dynamic adaptability, paving the way for reliable, real-time BP monitoring in diverse clinical and personal healthcare settings.

## Results
### BP measurement principle
For BP measurement, according to the theory of elastic mechanics[44,45], the exact BP value of a section of blood vessel can be calculated from the deformation and stiffness of the section of blood vessel. To satisfy the above theory, the following assumptions are made: (1) the blood vessel within the local area is uniform and its cross-section is circular; (2) the mechanical parameters of the blood vessel in the local area remain substantially unchanged (as illustrated in Fig. 1a).

In the following, we describe the underlying physical principles for calculating BP from local PWV and vascular diameter. BP refers to the force exerted by blood as it flows through blood vessels. Elastic arteries with high compliance can store greater volumes of blood, reducing the pressure impact on surrounding tissues, which in turn decreases PWV and subsequently lowers BP. Conversely, increased vascular stiffness reduces the pressure gradient and raises PWV, which may lead to elevated BP[25]. Therefore, BP can be characterized by quantifying the volume of blood passing through a specific vascular segment under stiffness conditions (see Methods and Supplementary Note 4 for details), expressed as[46,47]:

$$BP = \frac{PWV^2 \cdot \rho}{A \cdot \beta} + \theta = \frac{PWV^2 \cdot \mu}{A} + \theta \tag{1}$$

where $\beta$ is the beta stiffness related to the slope of pressure, $A$ is vascular cross-sectional area, $\rho$ is the blood density, $\mu = \rho / \beta$ represents the blood linear density defined as the rate of change in blood density along a unit length, and $\theta$ is linked to the intrinsic properties of the artery that maintain patency at low pressures.

From Eq. (1), accurately measuring BP requires obtaining physical parameters that represent vascular stiffness ($PWV$, $\beta$ or $\mu$) and deformation (cross-sectional area $A$). These parameters must be measured within the same local area to maintain the homogeneity of the vessel's material properties (e.g., Young's modulus and Poisson's ratio) and meet the uniformity assumption of the Bramwell-Hill equation[44]. Furthermore, the radial muscular artery is the preferred site for observing dynamic blood volume distribution within the vascular system[45], particularly when assessing linear blood density, as discussed later in this study.

Based on these considerations, we propose a measurement layout (Fig. 1b) in which the CSPM is positioned at a proximal or distal site along the vascular tree. The $PWV$ of the incident pulse transmission, reflective of the elastic properties of the local vascular segment, is determined from the arrival time difference recorded by two sets of PWF sensors (Pulse Waveform Sensor 1 and Pulse Waveform Sensor 2 in Fig. 1b). Simultaneously, the ultrasound sensing unit, located between the two sets of PWF sensors, measures the cross-sectional area (by capturing the vessel diameter), contributing to the $A$ parameter in Eq. (1). Building on the above theory, we developed a workflow for calibration-free blood pressure calculation, as illustrated in the flowchart presented in Supplementary Fig. 1.

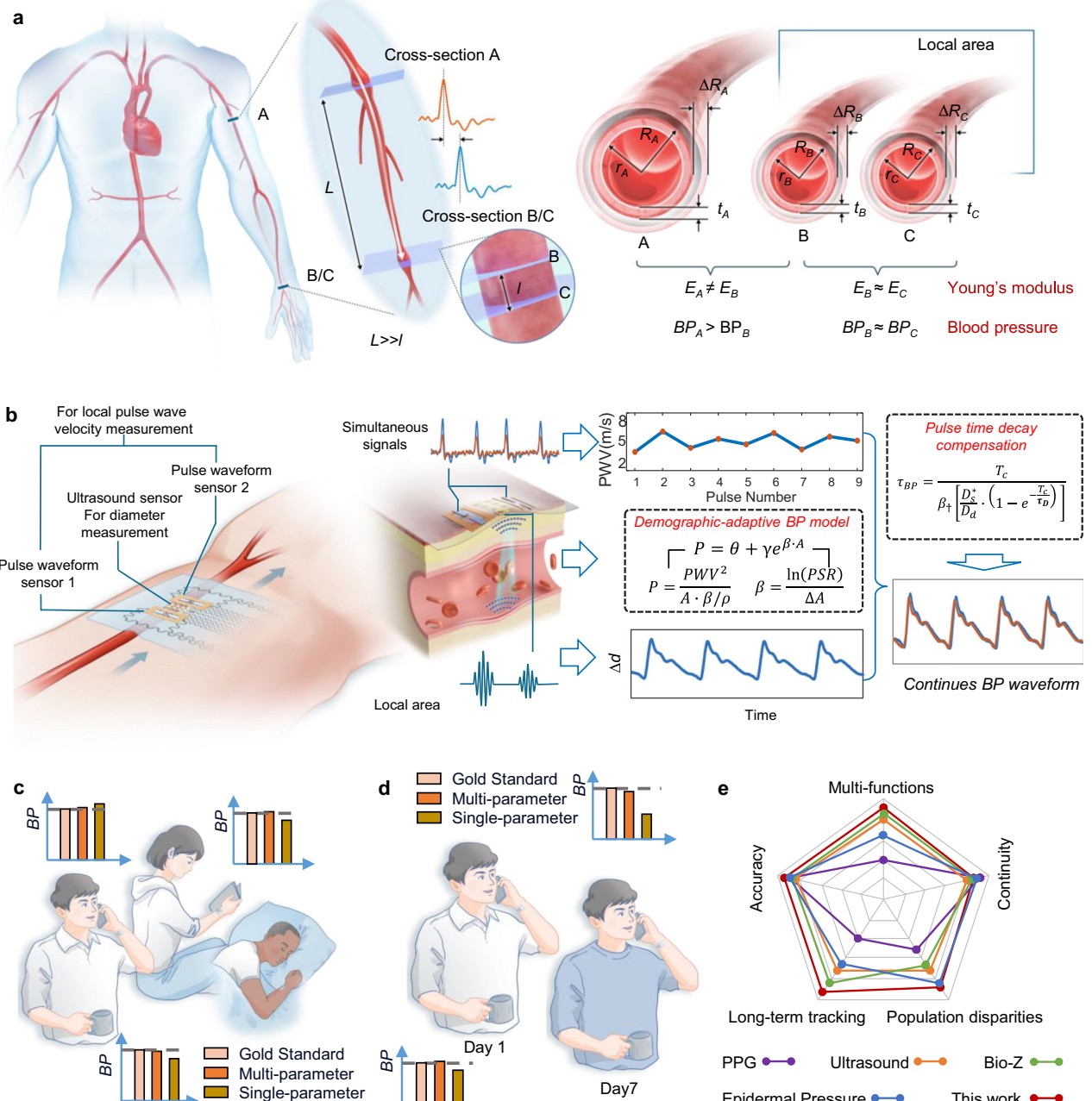

**Fig. 1 | Working principle and advantages of the CSPM. a** Schematic diagram of pulse wave propagation in a human artery and comparison of vascular morphology at different locations. **b** Working principle of the CSPM system, including device layout and algorithm. **c** Comparison of BP test results of different individuals (diverse gender, skin tone, and age) of gold standard, single-parameter and multi-parameter measurements without repeat calibration. **d** Comparison of BP test results of the same individual over 7 days of gold standard, single-parameter and multi-parameter measurements without repeat calibration. **e** Technical comparison between our device and published works utilizing photoplethysmography (PPG), ultrasound wall tracking, bioimpedance (Bio-Z), and epidermal pressure sensor for continuous BP monitoring in terms of multi-functions, accuracy, continuity, long-term tracking, and population disparities.

After establishing the foundational model, we further address the challenge of relative sliding between the sensor and the blood vessel during measurements — a frequently encountered issue that affects the performance of continuous BP tracking. Traditional approaches to this challenge primarily involve optimizing materials and structures by employing coordinated arrays of highly sensitive sensors to monitor vascular changes[30]. However, these methods face limitations due to pulse distortions caused by the finite resolution of the sensor patch and uncertain displacement disturbances during vessel pulsation, leading to degraded performance in continuous BP tracking[48]. To overcome these issues, we introduce a pulse time-decay compensation strategy (illustrated on the right side of Fig. 1b; details provided in Supplementary Note 5) that ensures consistent transformation of time constants extracted from the sensor pulse waveforms and BP waveforms within each heartbeat cycle. Importantly, the time decay parameter is dependent on BP and remains stable regardless of measurement position, even when pulsation amplitude varies. By integrating the pulse time-decay compensation strategy with the $\beta$ stiffness estimator in the CSPM, precise and continuous BP measurements can be reliably achieved.

As will be demonstrated in later sections, the CSPM and corresponding BP algorithm can record BP values across different

individuals (varying in gender, skin tone, and age) without the need for repeated calibration, while maintaining sufficient measurement accuracy (Mean Absolute Error, MAE=mean(abs(BP$_{ref}$-BP$_{est}$), SBP = 5.22 mmHg, DBP = 4.57 mmHg) (Fig. 1c). Furthermore, for the same individual, the method ensures accurate BP tracking over an extended period (up to 7 days) without degradation in device performance or algorithm accuracy (Fig. 1d). As such, the CSPM surpasses existing continuous BP monitoring devices in terms of multi-functionality, accuracy, continuity, long-term tracking capability, and its ability to accommodate population differences and other critical factors (Fig. 1e).

## Device design and fabrication

The BP calculation model underscores the importance of synchronizing vascular stiffness metrics (*PWV*, *β*, or *μ*) and vascular deformation (cross-sectional area *A*, *ΔA*) measurements within the same local area. To achieve this, we designed the CSPM layout as shown in Fig. 2a and Supplementary Fig. 2. The two PWF sensor sets are spaced 12 mm apart, a distance sufficient to differentiate their respective pulse signals. By analyzing these pulse waveforms, we can calculate the local *PWV* by dividing the distance by the time difference between the signals from the two PWF sensors. We chose a pressure sensor as the PWF sensor because of its high temporal resolution. Importantly, accurate *PWV* calculation requires PWF sensors with high sensitivity, cross-sensor uniformity, and rapid response. To meet these criteria, we selected laser-patterned piezoelectric thin-film materials for fabricating the PWF sensors. Piezoelectric materials convert mechanical energy, such as pressure, directly into electrical energy through the piezoelectric effect. This mechanism eliminates the need for an external power source, enabling piezoelectric sensors to function as autonomous, self-powered devices. The laser patterning approach offers several advantages, including fast response, high performance uniformity across devices, fabrication simplicity, and self-powering capability.

Depending on the mode of application, the PWF sensor is available in press-mode and self-adhesion mode (Fig. 2b). The self-adhesion PWF sensor applies minimal pressure to the skin and blood vessels, making it well-suited for unobtrusive measurement of local *PWV*. However, due to its lack of a backing support layer (such as a belt or cuff), the self-adhesion mode transmits a smaller pulse-generated force to the sensor. To enhance pulse signal detection, we incorporated a cantilever-based PWF sensor with a cavity structure, which offers heightened sensitivity by increasing the strain exerted on the piezoelectric element during vibration. Based on finite element analysis, the stress and deformation distributions of press mode and self-adhesion mode PWF sensors were analyzed, demonstrating that self-adhesion mode PWF sensors within a cavity exhibit greater deformation and strain compared to press mode sensors (Fig. 2c, Supplementary Note 2 and Supplementary Figs. 3, 4). This design not only improves sensitivity but also eliminates the need for additional straps or tapes that could otherwise compress and alter the blood vessels.

The ultrasound sensor is configured in a 2 × 4 transducer array, with each transducer individually controlled by eight stimulating electrodes positioned on the top silicon elastomer, along with a shared ground electrode at the base. This configuration enables precise vessel position mapping easily. Transducers are spaced at 1.6 mm to balance mapping precision with flexibility. The transducer is fabricated from a 1−3 composite material, which exhibits high electromechanical coupling coefficients in thickness mode—allowing for efficient conversion of electrical energy into vibrational energy—and a higher resonant frequency compared to traditional PZT materials with the same size, making it ideal for high-precision measurements (Supplementary Fig. 5). Additionally, the low acoustic impedance (15.3 MRayl) of the 1−3 composite enables improved acoustic coupling with soft biological tissues (-1.5 MRayl) compared to isotropic piezoelectric ceramic

materials (e.g., 33 MRayl for PZT). A copper cube serves as the vertical interconnect access (VIA), consolidating the upper and lower electrodes onto the same plane to facilitate electrical bonding (Supplementary Fig. 6). During operation, the ultrasound transducer is activated by a pulsed voltage signal, emitting ultrasound waves that penetrate tissues. When these waves encounter artery-tissue interfaces, part of the wave reflects back while the remainder continues forward into deeper tissue layers (Fig. 2d). The reflected wave, containing essential interface location information, is then captured by the same transducer (Fig. 2d). By analyzing time-of-flight (TOF) signals, the system captures pulsating vessel wall data, which can be converted into diameter change waveforms through amplitude peak shifts.

The CSPM is fabricated through a multi-step layered process using glass slides as temporary substrates. The PWF and ultrasound sensors are fabricated separately and later integrated through precise assembly (Supplementary Note 3 and Supplementary Figs. 7–9). Notably, the ultrasound sensor's ground electrode is directly bonded onto the transducer without a substrate, using transfer printing technology with a PDMS stamp (Fig. 2e). The height of the PDMS cavity in the PWF pressure sensors aligns precisely with the transducers, allowing a mortise-and-tenon-like assembly (Fig. 2f). The entire device is encapsulated in a low-stiffness silicone rubber (Ecoflex 00-30), which offers protection against sweat corrosion due to its hydrophobic properties. Additionally, the encapsulation material has a modulus similar to that of human tissue and excellent biocompatibility, as demonstrated by cell viability testing (Supplementary Fig. 10). The overall thickness of the CSPM is less than 450 µm (Supplementary Figs. 11, 12), and its total weight is under 1 g, minimizing any pressure on the skin, tissues, or blood vessels, thus reducing the risk of measurement errors from pressure-induced deformation. The device is stretchable and flexible, allowing it to be comfortably attached to human skin for long-term BP monitoring (Fig. 2g, h and Supplementary Figs. 13, 14).

## Device characterization

We first evaluated the performance of each sensing module within the CSPM. For the ultrasound sensor, which is tasked with precisely tracking vascular diameter and its fluctuations, we selected a piezoelectric transducer with a resonant frequency of 5 MHz to ensure high measurement accuracy and sufficient probe depth (Fig. 3a). Additionally, the 2 × 4 transducer array demonstrates consistent impedance characteristics, with an average electromechanical coupling coefficient of 0.51, indicating good uniformity in the ultrasound sensor (Fig. 3b and Supplementary Fig. 15). This uniformity enables efficient vessel mapping for rapid identification and diameter measurement without the need for additional manual adjustment. In order to effectively assess the condition of subcutaneous blood vessels, achieving adequate ultrasound penetration depth becomes imperative. The relative acoustic pressure (RAP) distribution of transducers with varying sizes was obtained utilizing the Matlab TAC_GUI toolbox (Fig. 3c). We selected a 1 mm transducer size to balance ultrasound penetration depth and device flexibility, surpassing subcutaneous blood vessel depths, with −20 dB as the pivotal threshold (Supplementary Figs. 16, 17). The ultrasound transducer's pulse-echo response was evaluated on an agarose-based hydrogel mimicking human tissue, revealing a narrow spatial pulse length (-1.32 µs) and a notable sensitivity of 31% within a −6 dB bandwidth (see Methods, Fig. 3d, Supplementary Fig. 18). Crosstalk phenomena among elements in the transducer array were examined at 5 MHz (see Methods and Supplementary Fig. 19), demonstrating stable crosstalk levels averaging around −57 dB, surpassing the common −30 dB threshold observed in the field[49]. To ensure the device achieves conformal contact with human skin, it is imperative for it to possess not only a thin profile but also a requisite level of stretchability. The device can be reversibly stretched up to 40% (Supplementary Fig. 20), which exceeds the maximum strain (about 20%) that human skin can withstand[50]. When

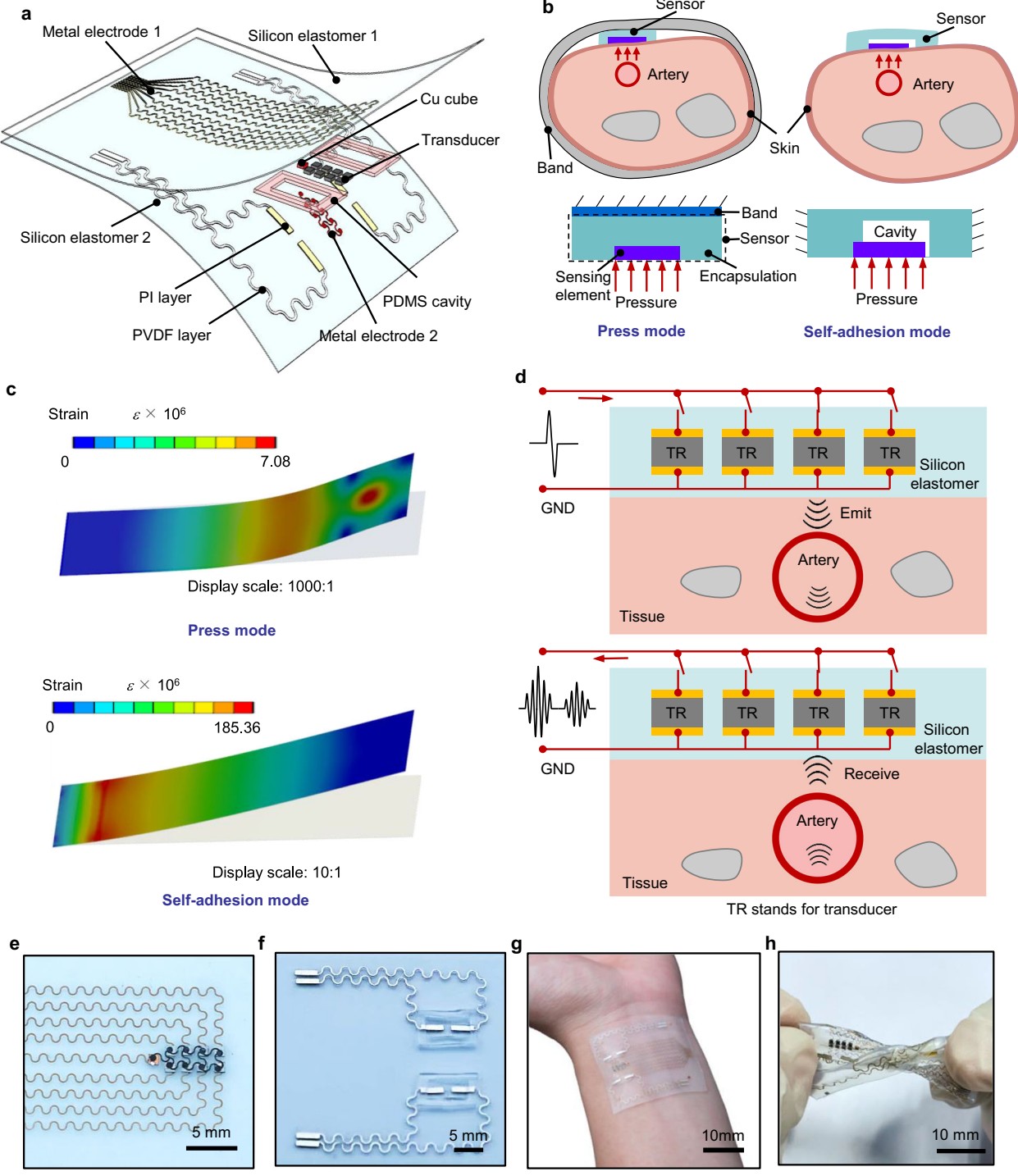

**Fig. 2 | Structure design of the CSPM. a** Schematic illustration of CSPM composite of PWF sensor and ultrasound sensor, which measures local pulse wave velocity (PWV) and diameter change of the same segment of artery for BP monitoring. **b** Two types of flexible PWF sensors: press mode using a belt or motor and self-adhesion mode using its own adhesive force. **c** Stain distribution of press mode and self-adhesion mode PWF sensors by finite element analyses (sensing part size: 1 mm × 4 mm), showing that the self-adhesion mode PWF sensor has larger stain than the press mode PWF sensor under the same load. **d** Working principle of ultrasound sensor including emitting ultrasound wave (up) and receiving ultrasound wave (down). **e** Optical image of ultrasound sensor. **f** Optical image of PWF sensor. **g** Optical image of CSPM attached on human skin. **h** Optical image of CSPM under twisting demonstrating the mechanical compliance and robustness of the device.

the device is stretched to 40%, the electrical impedances of the transducers remain almost consistent across different strain levels, providing evidence that the applied stretching (<40%) does not exert any discernible effect on the electrical properties of the device (Fig. 3e). Moreover, we measured the device's resonant frequency under cyclic stretching (Supplementary Fig. 21). The results show that the device maintains signal stability even after repeated strain of 40%,

thereby enhancing its mechanical durability in cyclic deformation applications.

For the PWF sensors, we observed that encapsulating the wearable sensor with viscoelastic silicone rubber reduces response speed. However, the cavity design effectively mitigates the impact of the silicone layer on the cantilever-type force sensor, resulting in a substantial improvement in the sensor's response time. According to step

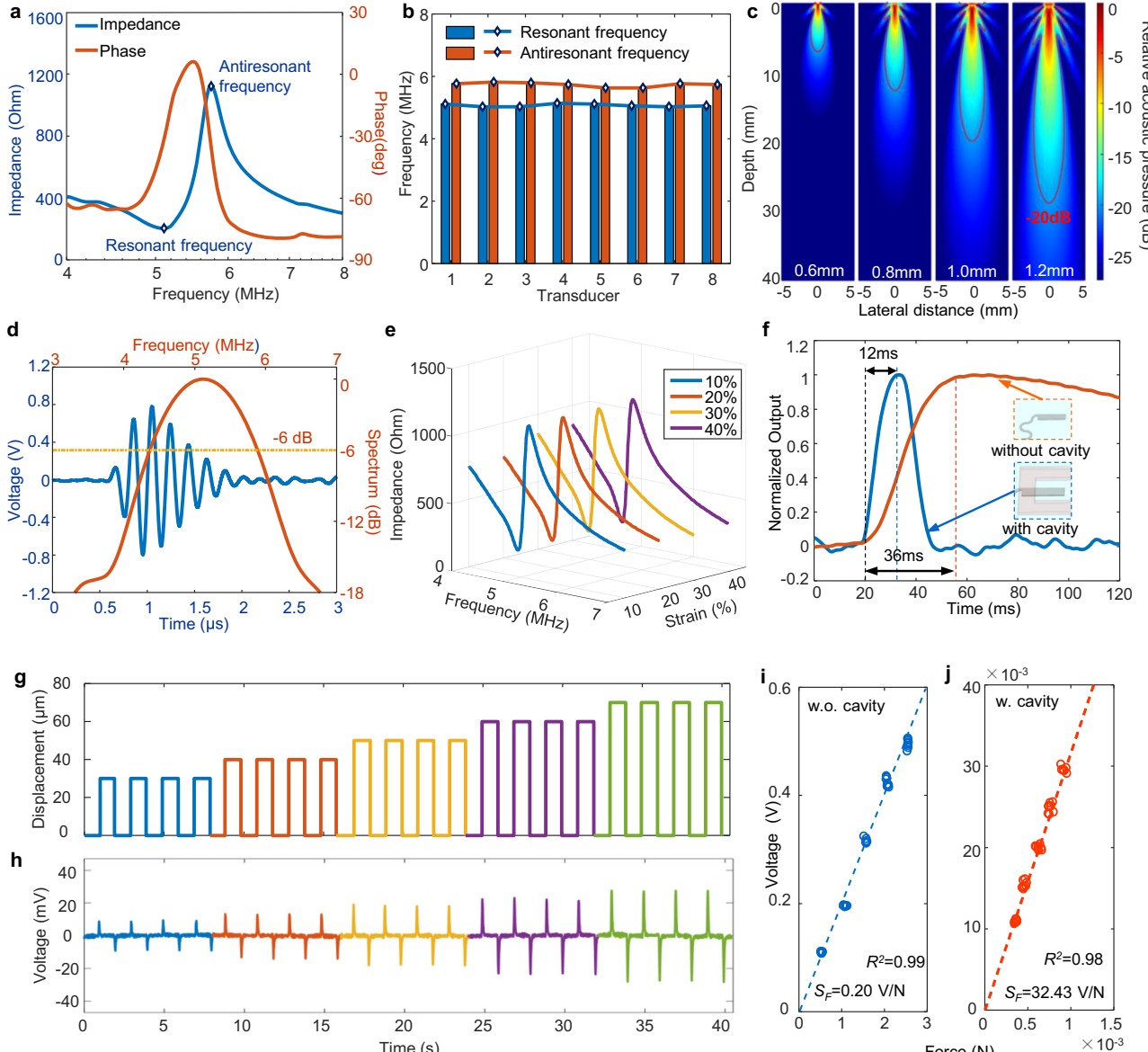

**Fig. 3 | Characterization of the CSPM. a** Impedance and phase angle spectra of the ultrasound transducer, showing good electromechanical coupling capability. **b** Resonance and antiresonance frequency variations of 8 transducer elements in CSPM. The mean values/SDs are 5.07 MHz/43.76 kHz (resonant) and 5.74 MHz/70.62 kHz (antiresonant), respectively. **c** Acoustic field simulation of ultrasound transducer with different sizes. **d** Pulse-echo response and frequency spectra, with a short spatial pulse length (-1.32 µs) with a peak to peak voltage of -1.6 V, and a central frequency of 5.11 MHz with good sensitivity of 31% at −6 dB bandwidth.

**e** Electrical impedances of ultrasound sensor under different strains. **f** Comparison of PWF sensor without and with cavity in terms of response time. **g** Square wave displacement loads with varying amplitudes applied on PWF sensor. **h** Square response of PWF sensor, showing a consistent trend with displacement loads. **i**, **j** Sensitivity test results for the stacked-type PWF sensor and cantilever-type PWF sensor, which are 0.20 V/N and 32.43 V/N, respectively, with coefficients of determination of 0.99 and 0.98 respectively.

response results (Fig. 3f), the response time of the cantilever-type PWF sensor is merely 1/3 of that observed in the stacked-type PWF sensor without cavity, indicating a considerable enhancement in responsiveness and faster signal detection capabilities for pulse waveform. Due to the fast response speed of the sensor, it adeptly tracks and records pulse waveforms in real-time synchronization (Supplementary Fig. 22). Applying different displacement loads to the PWF sensor leads to an output that grows with increasing displacement, demonstrating excellent linearity in their relationship (Fig. 3g, h).

The introduction of the cavity not only increases the response speed of the sensor, but also improves its sensitivity. According to the sensitivity test results for the stacked-type (press mode) PWF sensor and cantilever-type (self-adhesion mode) PWF sensor (Fig. 3i, j), the

stacked-type PWF sensor exhibits a force sensitivity of 0.20 V/N, while the cantilever-type sensor exhibits a notably higher sensitivity of 32.43 V/N, marking an -160-fold increase over the stacked-type sensor. To assess the sensitivity of various PWF sensors with distinct characteristic parameters, such as supporting layer thickness, comprehensive tests were conducted (Supplementary Fig. 23). Notably, the results reveal that the maximum force sensitivity is achieved when the cavity size measures 3 mm × 6 mm, and the supporting layer has a thickness of 30 µm (standard specifications for commercial materials). These results highlight the importance of optimizing the characteristic parameters to attain optimal force sensitivity in PWF sensors, offering improved detection and measurement capabilities for capturing and analyzing subtle variations in applied forces.

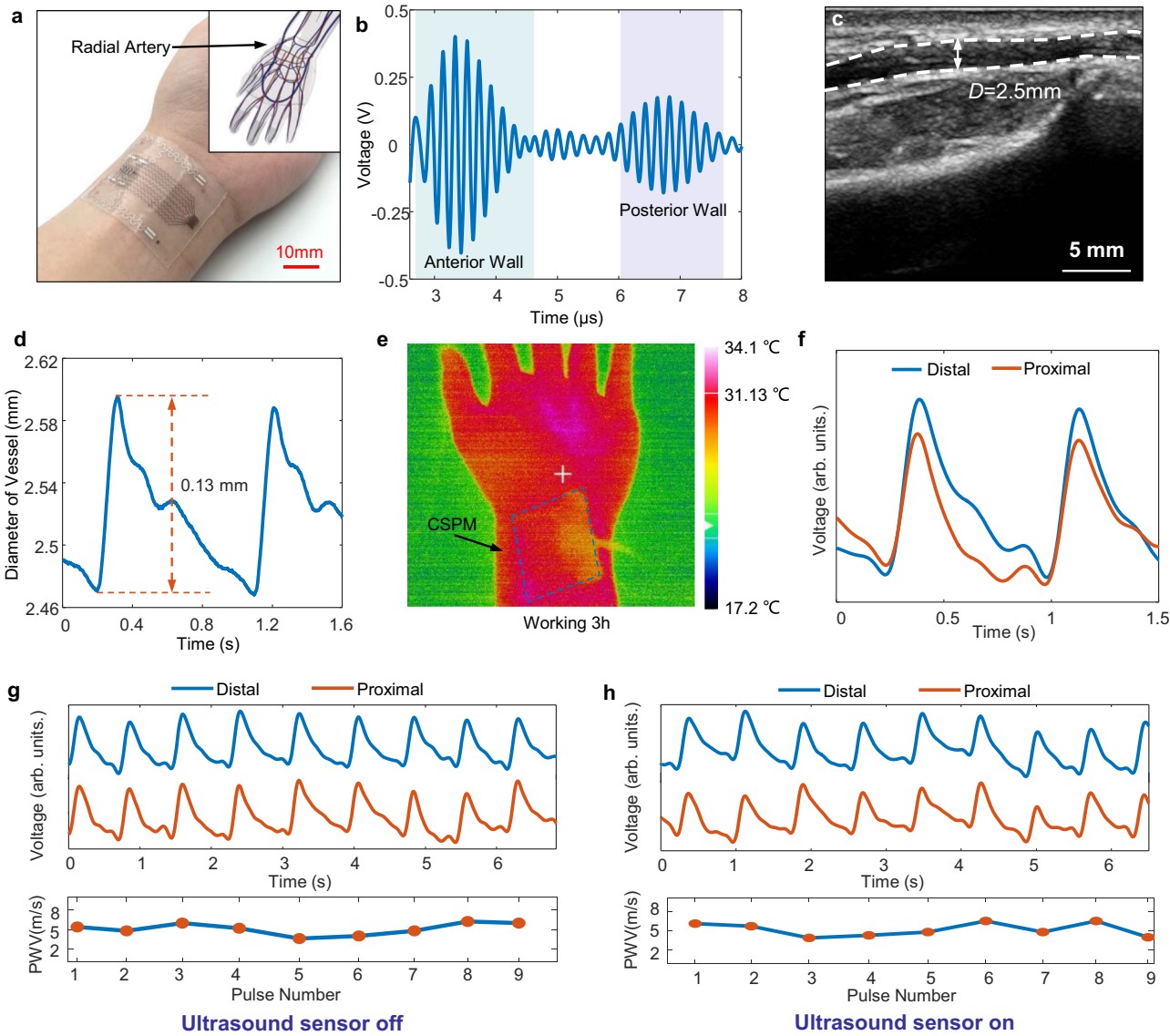

**Fig. 4 | Vascular parameter monitoring by the CSPM. a** Optical image of the SCPM on the human wrist for continuous measurements of radial pulse waveforms and local PWV. Inset shows the vascular structure of the testing site, adapted from https://simvascular.github.io/ (Copyright © SimVascular Development Team, 2023). **b** Received ultrasound echo signal of the radial artery with two clear echo peaks from the anterior and posterior walls. **c** Grayscale ultrasound image of blood vessels measured by commercial ultrasound imager. **d** Radial pulse waveforms of the blood vessel in real time with diameter change of ~0.13 mm. **e** Thermal distribution of the device after 3 h of operation, showing overall temperature remained largely stable. **f** Two pulse waveforms obtained by distal and proximal PWF sensors. **g** Pulse waveforms and the calculated local PWV when the ultrasound sensor is off. **h** Pulse waveforms and the calculated local PWV when the ultrasound sensor turns on.

## Performance validation

We proceeded to validate the performance of the CSPM system, confirming that it can attach conformally to human skin, even on areas with pronounced curvature, such as the wrist (Fig. 4a). The ultrasound echo signal of the artery was recorded, as shown in Fig. 4b, with distinct and easily identifiable peaks corresponding to the anterior and posterior walls of the artery. The diameter of the artery can be calculated based on the time difference between the two echo peaks. We also measured the diameter of the artery at the same location using a commercial ultrasound imager (Resona 7, Mindray), and the result is illustrated in Fig. 4c, which shows that the two measurement methods had similar measurement results, with the diameter measurement results being 2.5 mm and 2.53 mm, respectively. Our ultrasound sensor operates at a sampling frequency of 312.5 MHz, resulting in an effective resolution of 4.928 μm for tracing the arterial wall position. For the commercial ultrasound

imager, the resolution is approximately 68 μm based on the pixel size. Therefore, the resolution of the two devices may play a role in the differing measurement results. Moreover, our device operate at a high sampling frequency (~kHz), enabling the capture of detailed radial pulse waveforms (Fig. 4d). In contrast, the commercial ultrasound imager has a low imaging rate, which makes it unsuitable for radial pulse waveform extraction. When comparing the measurement results, we used a single snapshot of the diameter from the commercial imager and compared it with the diameter measured by our device at the half-maximum radial pulse intensity. As shown in Fig. 4d, our device tracks diameter variations between ~2.48 mm and 2.59 mm, and the commercial device's measurement falls within this range. The complete radial pulse waveforms of blood vessels can also be recorded in real time (Fig. 4d), enabling the calculation of $\Delta A$ for each cardiac cycle. Beyond the wrist, the ultrasound sensor also shows versatility in measuring blood vessel diameter at other

anatomical sites, such as the temporal and brachial arteries (Supplementary Figs. 24, 25).

Notably, our measurements reveal that the posterior wall of the artery exhibits a wider range of motion compared to the anterior wall for the radial and brachial arteries, while both walls of the temporal artery move by approximately the same amount. This discrepancy can be attributed to variations in vessel depth and diameter at different measurement locations. As a result, the device moves in sync with the vascular contraction and expansion, causing minimal variation in the distance between the device and the anterior wall of the blood vessel (Supplementary Figs. 26, 27). The results obtained using our device align closely with those obtained from commercial imaging systems, affirming its accuracy and reliability across multiple locations. Heating is a challenge faced by ultrasound devices, so we conducted temperature field distribution tests on developed the ultrasound sensor under long-term working conditions. The results indicate that, with the dual-mode system attached to the skin, after 3 h of operation, the overall temperature remained essentially unchanged (Fig. 4e and Supplementary Fig. 28). This means that the device can be applied to the human skin for extended periods of measurement without concerns about skin damage due to temperature elevation.

While measuring the artery diameter, the pulse waveforms are also recorded by two sets of PWF sensors (Fig. 4f), so that the local $PWV$ can be obtained by the two-point method from time-shifted blood pulses. Since the peak point of the pulse waveform is largely influenced by the propagation of reflected vascular pressure waves, we employ the first derivative maximum point as a reliable indicator for evaluating local $PWV$, which minimizes the impact of vascular wave reflections[51]. The 1st derivatives of the measured two pulse waveforms are illustrated in Supplementary Fig. 29, and the time difference between the two peak points is 3.41 ms. By recording continuous pulse waveforms (Fig. 4g), we can obtain local $PWV$ results, with a mean local $PWV$ of 5.11 m/s and a standard deviation of 0.91 m/s. Importantly, we compared the local $PWV$ measurements with and without the ultrasound sensor in operation, finding consistent results (Fig. 4g, h). This confirms that the ultrasound sensor does not interfere with $PWV$ measurement, allowing both types of sensors to operate simultaneously (Supplementary Movie 1).

## BP monitoring

Building on the multi-parameter physiological model outlined in Fig. 1 and Eq. (1), we now proceed to demonstrate how the CSPM enables demographic-adaptive, calibration-free, and cuffless BP monitoring. After extracting the $PWV$, vascular area ($A$), and its variation ($\Delta A$) during a cardiac cycle, the demographic vascular parameters—$PSR$, $\beta$, and $\mu$ — can be determined using Eqs. (2) and (3) (see Methods). Notably, the $cfPWV$ in Eq. (3) in Methods is not directly equivalent to the measured PWV from the CSPM ($PWV_{CSPM}$) due to differences in measurement sites. To relate these parameters, we established a linear relationship based on the equivalent $\beta$-based $\mu$ ($\mu_\beta = \rho/\beta$) and the reference $\mu$ ($\mu_{ref} = BP \cdot A/PWV^2$), where $\mu_{ref}$ is determined using reference $BP$, $A$, and $PWV$ values. Using commercial PWV test equipment and the CSPM, we obtained a linear relationship between pulse wave velocity (PWV) and PWV_CSPM (Supplementary Fig. 30): $cfPWV = k \cdot PWV_{CSPM} + c$. Given the anatomical location of the CSPM system, we infer that $PWV_{CSPM}$ can reliably represent the local $PWV$. To calculate $k$ and $c$, reference $BP$, $PWV$, and $A$ were measured from five subjects to obtain $\mu_{ref}$, $\beta_{ref}$ and subsequently $cfPWV$ using Eqs. (2) and (3) in Methods.

Before performing the BP algorithm, the basic demographic characteristics are determined during initial calibration. Supplementary Figs. 31, 32 show the personalized blood vessel cross-sectional area $A$, $PWV$, $\beta$ stiffness, and $\mu$ linear blood density of five subjects, where $\beta$ stiffness is calculated by Supplementary Eq. (6) and $\mu$ is calculated by substituting known blood pressure, $PWV$, and $A$ in Eq. (1).

The subject-specific linear blood density was estimated directly by substituting the $PWV$ and diameters in Supplementary Eq. (10), aligning well ($p < 0.0001$) with the reference linear blood density by substituting known blood pressure into Eq. (1). The calibration methodology presented herein focuses on the population parameter calibration, derived from and subsequently recalibrated using the dataset. This approach ensures that all model coefficients are determined from population-level information, thereby eliminating correlation with individual-specific data. This strategy offers the advantage of broad applicability across diverse populations, negating the need for subsequent recalibration. Specifically, for the coefficients outlined in Eq. (1), we employ population data for all calibration determinations ($n = 5$). The resulting linear fit yielded $k = 1.1307$ and $c = 1.3879$ (details in Supplementary Note 4). These constants were then applied during the test stage on new subjects to enable calibration-free BP measurement using Eqs. (1)–(3) in Methods.

The proposed methodology offers three key advantages: (i) Precise demographic vascular stiffness description: using Eq. (3) (see Methods) in conjunction with a large vascular parameter database[52], comprising 69 male and 97 female volunteers aged 22.3–83.2 years, we captured the $cfPWV$–$PSR$ relationship across the demographic population (Fig. 5a-i). (ii) Real-time vascular variation: As shown in Fig. 5a-ii, the parameters reflect real-time structural and functional changes in the artery caused by the interaction between the vascular wall and pulsatile blood volume in the local peripheral segment. For stiff arteries, a poor stiffness gradient leads to higher $PWV$, $\mu$, and vascular area, while elastic arteries exhibit the opposite. These variations are critical for accurate BP tracking. (iii) Time-decay compensation: The time-decay trend ($\tau$) of vascular diameter mirrors BP changes caused by vascular wall adjustments during each cardiac cycle (Fig. 5a-iii). Accordingly, a time-decay compensation strategy (Supplementary Note 5) is introduced to address structural distortions from minor slippage and localized shifts of the CSPM (Supplementary Figs. 33, 34), enhancing measurement accuracy. Together, these features enable calibration-free, reliable, and continuous BP monitoring for new users.

We first validated the performance of the time-decay compensation strategy in continuous BP measurement. Reference continuous BP data were collected using a commercial continuous blood pressure system (BIOPAC, MP160), while pulse waves and vascular diameter were simultaneously recorded by the CSPM from the same subject. The CSPM-generated continuous BP was calculated by inputting extracted parameters (i.e., $PWV$, $A$, $\Delta A$ and $\tau$) into the established model. Figure 5b, c illustrates the significant BP errors caused by vessel drift as detected by ultrasound[16] and the improvements achieved using the time-decay method in the CSPM (Supplementary Movie 1). This compensation strategy reduced the estimation error from -10 mmHg to 2 mmHg during continuous BP monitoring. Comparison of various blood pressure models in continuous blood pressure waveform tracking was carried out, demonstrating heightened accuracy (mean difference $\mu = -0.18$ mmHg, standard deviation $\sigma = 2.15$ mmHg) in BP assessment when employing the proposed BP algorithm, evidencing a notable reduction in measurement inaccuracies (Supplementary Figs. 35–37).

Regarding manufacturing reproducibility of the device design, we fabricated three CSPMs using the same method and conducted BP measurements (Supplementary Fig. 38). These three devices can effectively measure continuous BP waveform. This demonstrates that our device design and proposed fabrication method have good manufacturing repeatability. To evaluate the impact of sweat, we tested the device's performance in both the absence and presence of sweat. The BP measurement results are shown in Supplementary Fig. 39, demonstrating minimal sweat-induced damage to the device. This is because the device is encapsulated in silicone rubber, whose hydrophobic properties effectively prevent sweat corrosion.

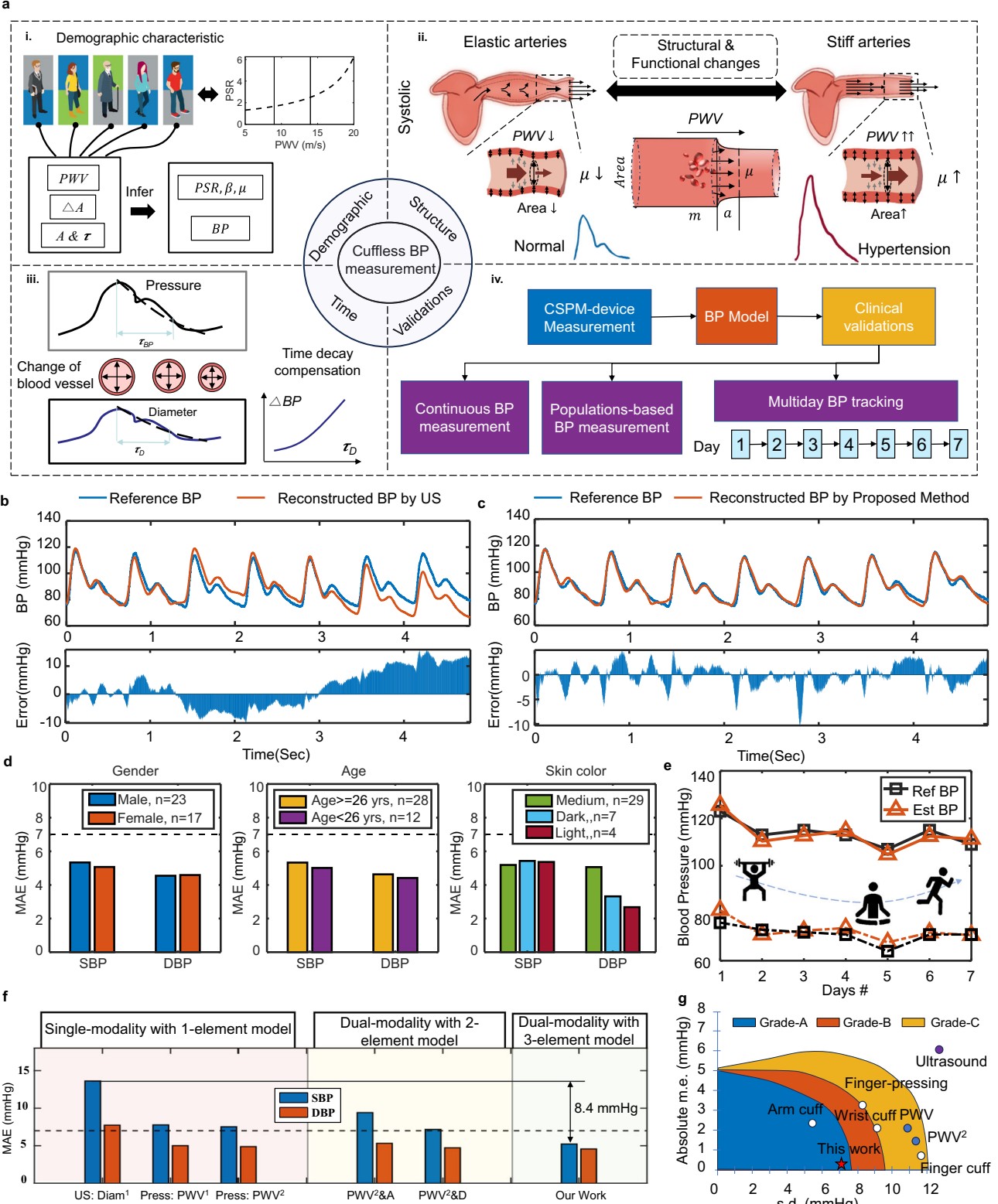

We next evaluated the CSPM's performance in a diverse sub-population by comparing it to a commercial device for snapshot BP measurements. This study included 45 subjects representing a variety of skin tones, genders, and age groups, as well as different measurement scenarios (continuous and snapshot BP readings). Data from the first five subjects, used to align PWV and carotid-femoral pulse wave velocity, were excluded, leaving the remaining participants as new users for CSPM validation without any calibration.

Figure 5d shows CSPM performance across gender, age, and skin tone subpopulations. Similar BP errors were observed in males (MAE: SBP = 5.34 mmHg, DBP = 4.55 mmHg, $n = 23$) and females (MAE: SBP = 5.07 mmHg, DBP = 4.59 mmHg, $n = 17$). These subtle discrepancies in error may stem from intrinsic variations in vascular parameters between the sexes, yet remain accurately measured by our CSPM. Similarly, younger participants (<26 years) exhibited slightly larger errors (MAE: SBP = 5.33 mmHg, DBP = 4.63 mmHg, $n = 12$) compared to

**Fig. 5 | BP measurement algorithm and measurement results. a** Framework of the CSPM-BP measurement. **a-i.** The demographic vascular information (e.g., $PSR$, $\beta$ and $\mu$) can be determined by Eqs. (2) and (3) based on the demographic adaptive module. **a-ii.** The vascular parameters comparison between the normal and hypertensive. **a-iii.** Consistency of the pulse-based time-decay can compensate the vessel drifting effect during continuous BP measurement. **a-iv.** Validations of BP measurement. **b, c** The comparison of continuous BP measurement under the traditional ultrasound (US)-based and the proposed time-decay compensation strategy, where the maximum error is -10 mmHg during the BP measurement in (**b**). **d** Mean absolute error (MAE) of measured BP across diverse subpopulations, encompassing gender, age, and skin tone, which verifies the effectiveness of the

proposed BP model. **e** Long-term (7 days) BP monitoring results for the same subject with various daily activities (Day 1, weight lifting 10 min; Day 2–5, sitting 10 min; Day 6, running 10 min; Day 7, sitting 10 min). **f** Comparison of the proposed CSPM-demographic adaptive method with other BP measurement methods, including diameter-based method, PWV-based method, $PWV^2$-based method, $PWV^2$-diameter method and $PWV^2$-area method. **g** Comparison of our work with other BP measurement methods in terms of absolute mean error (m.e.) and standard deviation (s.d.) of BP measurement under IEEE-2014/2019 accuracy requirements, including arm cuff, finger-pressing, wrist cuff, finger cuff, ultrasound, PWV-based method and $PWV^2$-based method.

older participants (≥26 years; MAE: SBP = 5.01 mmHg, DBP = 4.41 mmHg, $n$ = 28). Across skin tones, consistent low BP errors were observed: light (MAE: SBP = 5.36 mmHg, DBP = 2.68 mmHg, $n$ = 4), dark (MAE: SBP = 5.43 mmHg, DBP = 3.32 mmHg, $n$ = 7), and medium (MAE: SBP = 5.19 mmHg, DBP = 5.06 mmHg, $n$ = 29). Theoretically, skin tones should not impact test results as our device operates on mechanical principles (pressure and ultrasound) rather than optical measurements. The results show similar SBP measurement errors across the three skin tones, while some discrepancies exist in DBP measurements. These findings demonstrate that subject-specific vascular stiffness parameters effectively represent BP across diverse populations (additional statistical comparisons are provided in Supplementary Tables 2, 3).

To further evaluate the performance of our method, we conducted additional analyses by comparing the BP measurement results with and without calibration. Here, calibration involves using the BP data acquired from each individual to calibrate the constants in our model. For these analyses, the reference BP values were obtained using a commercial sphygmomanometer. First, we calculated the average mean error (m.e. = mean($BP_{ref}$-$BP_{est}$)) and standard deviation (s.d. = std($BP_{ref}$-$BP_{est}$)) of BP measurements with individual calibration for each subject, as presented in Supplementary Table 4. From the Bland–Altman plots in Supplementary Fig. 40, the average m.e. and s.d. were obtained as 0.52 ± 3.38 mmHg for SBP and 0.88 ± 2.75 mmHg for DBP. For measurements without individual calibration, the results for all subjects are presented in Supplementary Table 5. From the Bland–Altman plots in Supplementary Fig. 41, the average m.e. and s.d. were obtained as 0.90 ± 7.14 mmHg for SBP and 1.77 ± 5.51 mmHg for DBP.

Additionally, we used the original CSPM devices to measure BP in a group of 20 subjects ~6 months after their fabrication. These results were compared to those obtained drom the initial 25-subject test group, who were measured using the same batch of devices. The mean error of BP across this 6-month time window is presented in Supplementary Fig. 42. The results confirm the device's reliability in measuring blood pressure for 45 subjects over six months.

Further demonstration of individual long-term BP monitoring is presented in Fig. 5e, which shows 7-day BP measurements during various daily activities, compared with commercial snapshot BP readings (HEM-FL31, OMRON) which is an FDA validated cuff-based device. Data from the first day were used to fine-tune the CSPM system's vascular parameters, enabling more accurate BP tracking. Over the monitoring period, the SBP errors of the long-term monitoring exhibited a bias of −0.458 ± 2.308 mmHg, while the DBP errors showed a bias of 1.551 ± 1.58 mmHg (Supplementary Table 6), both of which meet the ANSI/AAMI/ISO 81060-2 accuracy requirements for intermittent BP measurement[53–55]. Similar results were also obtained from another subject's long-term blood pressure tracking without repeated calibration (Supplementary Fig. 43 and Supplementary Table 7).

Finally, a performance comparison of the demographic-adaptive CSPM method with traditional ultrasound diameter-based cuffless BP measurements and established hybrid PWV-based methods reveals a significant reduction in both SBP and DBP estimation errors[13,24,56]

(Fig. 5f). In addition, we conducted a quantitative comparison of our method with existing blood pressure measurement techniques in terms of time complexity, spatial complexity, degree of complexity, and population accuracy, and so on (Supplementary Table 8). The results demonstrate that while our method shows a slightly higher time complexity, it significantly outperforms other methods in terms of accuracy in population. In terms of space complexity, our method remains comparable to existing methods, ensuring its practicality and feasibility. The CSPM and corresponding BP algorithm achieved extremely high measurement accuracy under the BHS protocol (MAE: SBP = 5.22 mmHg, DBP = 4.57 mmHg), comparable to arm-based cuff methods (Fig. 5g). We conducted a comparative analysis with similar works, as outlined in Supplementary Table 9. Our device and BP algorithm achieve significant advancements in enhanced cardiovascular insights, BP tracking without calibration, and adaptability across diverse demographics, while maintaining high BP detection accuracy and reliable continuous monitoring.

## Discussion

We have developed a conformal and stretchable dual-mode piezoelectric microsystem, combined with a demographic-adaptive BP algorithm, to enable calibration-free continuous BP tracking with accuracy comparable to professional medical devices, even during long-term monitoring. By leveraging an integrated design strategy and advanced microfabrication techniques, the device simultaneously detects hemodynamic and structural parameters of individual blood vessels in the same local area, including local $PWV$ and vessel diameter. The BP algorithm incorporates dynamic demographic feature learning based on bimodal data, ensuring broad applicability across diverse populations. Additionally, a time-decay compensation approach mitigates measurement discrepancies across multiple readings and user groups, enhancing reliability in long-term BP monitoring without frequent recalibration. The simultaneous acquisition of signals at the same local position strengthens the correlation between these signals for the BP model and algorithm, revealing the intricate relationships between various physiological parameters and BP. This capability, along with the relaxed calibration requirements, underscores the clinical significance of the proposed CSPM system for long-term continuous BP monitoring across diverse populations.

Building on our current findings, future work can focus on further enhancing the performance of the device and algorithm. For instance, integrating additional sensors to measure more vascular parameters (e.g., blood density) may enable more comprehensive data acquisition, uncovering deeper correlations between these parameters and BP. This expansion could lead to the development of a more precise and adaptable BP model. Simultaneously, leveraging cloud platforms to collect and analyze extensive experimental data would provide valuable insights into the complex relationships between physiological parameters, demographics, and disease-specific factors, further refining the BP calculation algorithm. Ultimately, these advancements aim to achieve fully calibration-free BP measurement, opening the door to widespread adoption in both clinical and at-home settings.

Our research currently relies on calibration data from a relatively small cohort of five participants to establish relationships between parameters such as PWV, β stiffness, and BP. The limited sample size poses potential limitations to the generalizability of our findings. The current study's 5-sample calibration size is suitable for small populations (dozens to hundreds of individuals) but insufficient for larger, more diverse ones. A small sample may miss extensive physiological variations in larger groups, undermining the algorithm's robustness, reducing prediction accuracy, and causing overfitting—making established correlations perform poorly on new individuals with different characteristics. Thus, matching the calibration sample size to the test population scale is critical, as a more appropriate size boosts the algorithm's ability to handle variability and ensures reliable, accurate results.

Since small samples may not represent larger, diverse populations, calibration sample selection requires care. Instead of random sampling, samples should be deliberately chosen to cover a wide range of parameters (e.g., PWV, β stiffness, BP). This targeted approach helps the calibration process account for full physiological differences, improving the method's generalizability, capturing complex parameter relationships, and enhancing the model's adaptability to real-world scenarios.

For sufficiently large populations, using a limited number of individuals (not the entire group) for parameter calibration is an emerging trend in future large-cohort blood pressure measurement. Subsequent research will focus on exploring how to conduct model calibration with the minimum number of individuals while maintaining algorithm stability, involving the development of innovative sampling strategies, optimization techniques, and validation methods to balance sample efficiency and algorithm performance.

## Methods

### Details of BP calculation principle

While PWV and A can be measured directly, other parameters in Eq. (1) are obtained indirectly. The stiffness parameter β in Eq. (1) is an intrinsic characteristic of the blood vessel wall that indicates how vascular stiffness adjusts relative to changes in BP and cross-sectional area. Clinical observations have shown that pulsatile vascular stiffness varies with BP levels and correlates demographically with PWV[52]. Consequently, once PWV and A are determined by the CSPM sensors, β can be estimated using the following equations:

$$\beta = \frac{\ln(PSR)}{\Delta A} \tag{2}$$

$$PSR = \frac{1}{1 - (a \cdot cfPWV + b)} \tag{3}$$

where $\Delta A$ is the maximum change in cross-sectional area during one cardiac cycle, PSR is the pulse stiffening ratio[46], and $a = 0.04$ and $b = 0.04$ are demographic constants that can be derived from clinical data[52] (see Supplementary Note 4 for further details). cfPWV is the carotid-femoral pulse wave velocity. Note that we aligned the cfPWV with our measured PWV through a linear mapping of β-based $\mu$ ($\mu_\beta$) to a reference $\mu$ derived from Eq. (1), and details are provided in the Results section. Additionally, since blood density $\rho$ shows considerably less variation than β stiffness in the general population (see Supplementary Table 1), the $\mu_\beta$ can be calculated given a known β under a fixed $\rho$, with $\rho$ obtained directly from the literature[57].

### Fabrication of ultrasound sensor

A laminate of Cu/polyimide(PI) (18 μm/12.5 μm) film is applied onto the PDMS/glass substrate, followed by the patterning of electrodes using a UV laser cutter (Protolaser U4, LPKF). After the removal of excess material, the electrode pattern is retrieved from the PDMS-coated glass slide by water-soluble tape. The transfer-printing technique is employed to transfer the bottom electrode pattern onto an Ecoflex 00-30 (Smooth-On, Inc.) substrate spin-coated on a dextran-decorated glass slide, and transfer the top electrode pattern onto a PDMS/glass substrate. The piezoelectric transducers, along with a Cu cube for VIA, are bonded onto the bottom electrode using low-temperature solder paste (Sn42Bi57.6Ag0.4 (melting point, 138 °C), ALPHA Inc.). Alignment of the top electrode on the PDMS/glass substrate with the piezoelectric transducers is accomplished by referencing markers, facilitated by a microvisual system. Finally, the top electrode is bonded to the transducers using low-temperature solder paste, followed by the removal of the PDMS-coated glass slide to form the ultrasound sensor (Device 1).

### Fabrication of PWF sensor

The fabrication process of PWF sensor begins with laminating a polyvinylidene difluoride (PVDF) film (TE Connectivity, 28 μm) onto a PDMS-coated glass slide. The PWF sensor pattern is then defined by a laser cutter. After removing the excess PVDF material, the PWF sensor pattern is transferred printed onto an Ecoflex 00-30 substrate that has been spin-coated onto a dextran-decorated glass slide. A PDMS cavity is bonded to the PWF sensor using uncured PDMS as an adhesive to form the PWF sensor (Device 2).

### Integration of dual-mode CSPM

The PWF sensor (Device 2) is aligned with the ultrasound sensor (Device 1) through markers using a microscopic visual system, and then Device 2 is placed onto the substrate of the device 1 like a mortise and tenon structure. The integrated system was placed at a 45-degree angle and uncured Ecoflex 00-30 is poured over it. Once the Ecoflex 00-30 cures, the glass slides are peeled off to form the dual-mode CSPM. Finally, an additional layer of Silbione is blade-coated on the Ecoflex substrate as the adhesion layer.

### Impedance and phase angle testing of ultrasound transducer

An impedance analyzer (IM7581, HIOKI) is adopted to measure the impedance and phase angles of the ultrasound transducer. The scanning range is set to 4–8 MHz. The electromechanical coupling coefficient is a crucial parameter to characterize the ultrasound transducer's ability to efficiently convert between electrical and mechanical energy forms. The electromechanical coupling coefficients ($k_t$) of the ultrasound transducer can be derived by

$$k_t = \sqrt{\frac{\pi f_r}{2 f_a} \tan\left(\frac{\pi}{2} \frac{f_a - f_r}{f_a}\right)} \tag{4}$$

where $f_r$ and $f_a$ are the resonant frequency and antiresonant frequency, respectively, which can be obtained from the impedance and phase angle spectra.

### Pulse-echo response testing of ultrasound transducer

In order to test the pulse-echo response of the ultrasound transducer, an experimental system was constructed, including a pulser-receiver (CTS-8077PR, Goworld Corp.), an oscilloscope (Picoscope 6404E, Pico Technology), and a copper (Cu) sheet embedded in Agarose as the reflector. The data were collected by using NI labview 2018, and analyzed by using Matlab R2024b. The acoustic parameters of the Agarose are similar to those of human tissue. The frequency bandwidth (BW) of the echo signal at −6 dB is determined by

$$BW = \frac{f_u - f_l}{f_c} \times 100\% \tag{5}$$

where $f_u$ is the upper frequency, $f_l$ is the lower frequency, and $f_c$ is the central frequency.

## Fabrication of Cu sheet-Agarose model

Agarose powder was mixed with water at a w/v ratio of 1%. The mixture was heated to boiling and the Agarose powder was completely dissolved in the water. After cooling to about 60 °C, the mixture was poured into a mold with a copper sheet. After cooling, the mold was demolded to obtain the Cu sheet-Agarose model.

## Crosstalk testing of ultrasound transducer array

In order to evaluate the crosstalk of ultrasound transducer array, an experimental system was constructed, including a pulser-receiver, an oscilloscope, and a copper (Cu) sheet embedded in Agarose as the reflector. The testing method is setting one piezoelectric transducer in excitation mode and the remaining seven transducers in reception mode in turn. The decibels (dB) in crosstalk between ultrasound transducers is determined by

$$Crosstalk(dB) = 20 \ \log 10 \left( \frac{V}{V_{ref}} \right) \tag{6}$$

where $V_{ref}$ is the reference voltage, and $V$ is the measured voltage.

## Mechanical testing of ultrasound sensor

The mechanical evaluation of the ultrasound sensor necessitates a custom-made bearing-type rail tensile tester. Strain is calculated by dividing the tester's extension by the distance between two clamping claws. Strain levels are set at 10%, 20%, 30%, and 40%. Images capturing the device at various strain levels are acquired using an optical microscope. Following the device's elongation, the impedance of the ultrasound sensor is assessed using an impedance analyzer.

## Calibration of PWF sensor

The experimental setup of PWF sensor calibration involves a suite of equipment, including a 3-axis manual motion system, a commercial pressure sensor (9119AA1, Kistler), a pressure sensor amplifier (Type 5080, Kistler), a piezo actuator and its controller (NPC3, Newport), a current amplifier for the PWF sensor (SR570, Stanford Research Systems), and an oscilloscope (Picoscope 6404E, Pico Technology) for data acquisition. All the devices are mounted on an optical table to mitigate external disturbances and enhance experimental precision. Initially, the PWF sensor is positioned onto the commercial pressure sensor. Subsequently, the manual motion platform is meticulously adjusted to establish precise contact between the piezo actuator and the PWF sensor. The actuator is then manipulated to produce varying displacements, exerting force on the sensor. Finally, the oscilloscope and data acquisition system capture and record the output signals from both the PWF sensor and commercial pressure sensor, facilitating subsequent analysis and assessment.

## ECG correlation with PWF sensor

To investigate the correlation between ECG signals and the PWF sensor, the experimental setup comprises biometric measurement equipment (MP160, Biopac), an amplifier (YE5853, Jiangsu Lianneng), and an oscilloscope for data acquisition. ECG electrodes are positioned on the arm and linked to the biometric measurement equipment, while the PWF sensor is affixed to the wrist. Both ECG and PWF sensor signals are concurrently recorded and analyzed to explore potential correlations. During the test, the subjects remained seated and maintained a consistent physiological and psychological state to ensure the stability of the hemodynamic signals.

## Temperature field testing of CSPM

The experimental arrangement for temperature field testing comprises the application of a thermography device (Ti480P, Fluke) and a pulser-receiver. To commence, activate the ultrasound sensor affixed to the skin in excitation mode. Subsequently, conduct thermal imaging using the thermography device at the onset of the experiment and at 1-h, 2-h, and 3-h intervals thereafter.

## Crosstalk testing of CSPM

In order to evaluate the crosstalk of the dual-mode device, an experimental system was constructed, including a pulser-receiver, an oscilloscope and an amplifier. The PWF sensor in the dual-mode microsystem functions as a piezoelectric sensor and operates independently without requiring a power source, so it does not interfere with the ultrasound sensor. Therefore, the focus lies solely on examining the crosstalk from the ultrasound sensor to the PWF sensor. During the experiment, the dual-mode microsystem is initially attached to the radial artery of the wrist. Subsequently, measurements are taken from the PWF sensor concurrently with the ultrasound sensor operating in excitation mode and during its inactive state. The mutual crosstalk between these two sensor types within the dual-mode microsystem is analyzed based on the collected data.

## Cell viability assay of CSPM

L929 cells (purchased from ATCC, CCL-1) were inoculated onto the dual-mode device surface and cultured for 24 h. Dilute calcein-AM and ethidium homodimer-1 according to the manufacturer's instructions to create the staining solution. Typically, calcein-AM is used to stain live cells (green fluorescence), and ethidium homodimer-1 is used to stain dead cells (red fluorescence). The culture medium was removed from the cells, and the cells were gently washed with PBS to remove any residual medium. Add the staining solution to the cells, ensuring they are fully covered. Incubate the cells with the staining solution for the recommended time, usually around 15–30 min at room temperature or 37 °C, depending on the kit instructions. After incubation, gently wash the cells with PBS to remove excess dye. Place the dual-mode device under a fluorescence microscope. Use appropriate filters to visualize live cells (green) and dead cells (red). Capture images for analysis. Analyze the images to determine the ratio of live to dead cells. Depending on the software available, the fluorescence intensity or count the number of live and dead cells was quantified.

## Calibration of CSPM

For the calibration of CSPM, the test data from only 5 subjects is needed. During the BP measurement, subjects are required to maintain a seated position with the cuff of a commercial sphygmomanometer (HEM-FL31, OMRON) wrapped around the left upper arm, while the CSPM is worn on the wrist of the same side. The sensor is manually positioned on the skin. The procedure is as follows: Initially, the approximate location of the radial artery is identified by finger sensing the pulse on the wrist. The sensor array is then brought into contact with the identified location, and the program begins tracking ultrasound signals from the sensor pixels. The array is gently and slowly slid perpendicular to the artery. When a maximal shift between two ultrasound echo peaks (corresponding to the anterior and posterior walls of the artery) is observed, the sensor is correctly positioned above the artery. Once the correct position is identified, the whole sensor patch is gently pressed onto the skin to fix it in place. The SBP and DBP values are measured using the commercial sphygmomanometer, while the CSPM is adopted to measure the diameter of the artery and the local PWV. To ensure consistency in testing, all measurements were taken 2–3 h after eating. During measurement, subjects were instructed to sit quietly and rest for 10 min before testing to avoid BP fluctuations caused by movement, and the device automatically performed a calibration process to ensure accuracy. There was minimal arm movement during the experimental testing process to avoid significant motion artifacts, and the wrist was placed in front of the heart and at a consistent height. Each experiment lasted ~30 min. All human experiments were conducted in accordance with the protocol approved by the Hong Kong Science and Technology Park

(HKSTP) Clinical Research Ethics Committee and followed guidelines. The consent forms were signed by the subjects prior to the experiments.

When the CSPM was working, the ultrasound sensor was activated by a pulser-receiver with a peak voltage of 200 V, using the transmit/receive mode. The pulse repetition frequency was set at 1000 Hz. The echo signal was received by an oscilloscope with a sampling frequency of 312.5 MHz. Signal analysis based on the ultrasound flight time allows for the calculation of propagation distance by multiplying the ultrasound speed in the medium, which is assumed to be constant in human tissue. Subsequently, the diameter of the target artery is calculated. Additionally, the signal from the PWF sensor is captured using a data acquisition card, with the sampling frequency configured at 315 kHz. This setting is crucial to differentiate between the two pulse waves effectively and uphold the measurement accuracy of the local PWV. Finally, the CSPM is calibrated by the BP data measured by the commercial sphygmomanometer according to the developed BP estimation model in Supplementary Notes 4 and 5.

### Validation of BP recording across different individuals

During the validation of BP recording across different individuals, 45 subjects of different genders and ages participated in the experiment. In the discrete BP test, the subjects were required to remain in a sitting position, and the cuff of a commercial sphygmomanometer (HEM-FL31, OMRON) was wrapped around the left upper arm. In the continuous BP test, a continuous BP system (MP160, BIOPAC) was used instead of the commercial sphygmomanometer, and the BP acquisition frequency of BIOPAC was set to 2000 Hz. While measuring BP with the commercial device, the CSPM was worn on the wrist on the same side, and the measured data were converted into BP values according to the calibration results. For the 45 subjects wearing the CSPM, the positioning was completed independently by 3 operators, each of whom had undergone basic training.To maintain the consistency of the test, all BP measurements were tested 2–3 h after eating. During measurement, subjects were instructed to sit and rest for 10 min before the test to avoid BP fluctuations caused by exercise. There was almost no arm movement during the experimental test to avoid obvious motion artifacts, and the wrist was placed in front of the heart and at a consistent height. Each experiment lasted about 30 min.

### Validation of BP recording over an extended period

The long-term accuracy of the calibrated CSPM was evaluated in two healthy subjects. BP measurements from the calibrated CSPM and commercial sphygmomanometer (HEM-FL31, OMRON) were tracked for 7 days during daily measurements. The average differences in SBP and DBP measurements from our device and sphygmomanometer were calculated for each measurement day. To maintain the consistency of the tests, all BP measurements were taken 2–3 h after eating. Both subjects engaged in daily activities, including exercise (weight lifting), sitting, and running. Before the test, one subject randomly performed daily activities (Day 1, weight lifting 10 min; Day 2–5, sitting 10 min; Day 6, running 10 min; Day 7, sitting 10 min), while the other subject remained seated for 10 min before each test. There was little arm movement during the experimental test to avoid obvious motion artifacts, and the wrist was placed in front of the heart and at a consistent height. Each experiment lasted ~30 min.

### Ethics

Every experiment involving animals, human participants, or clinical samples has been carried out following a protocol approved by an ethical commission. Each participant gave informed written consent.

### Reporting summary

Further information on research design is available in the Nature Portfolio Reporting Summary linked to this article.

## Data availability

All data supporting the findings of this study are available within the article and its supplementary files. Any additional requests for information can be directed to, and will be fulfilled by, the corresponding authors. Source data for the figures are available from Figshare with the identifier https://doi.org/10.6084/m9.figshare.30451109. Source data are provided with this paper.

## Code availability

The algorithms and all parameters used for analysis and calculation are available in Methods and Supplementary Information.

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

## Acknowledgements

This project was supported by the General Research Fund (RGC Ref No. 14210124, N.Z.) from the Research Grants Council of Hong Kong, C-type Project from Science, Technology and Innovation Commission of Shenzhen Municipality (Reference No. 202205303000149, N.Z.), National Natural Science Foundation of China (Grant no. 52305611, C.L.) and Hong Kong Center for Cerebro-cardiovascular Health Engineering (COCHE) under the InnoHK Scheme of Hong Kong SAR.

## Author contributions

S.Q., C.L., and N.Z. conceived the idea. C.L., Z.J., and L.Z. designed, fabricated, integrated, and tested the devices. S.Q. derived and developed the BP model and algorithms. C.L., Z.J., S.Q., and N.Z. designed the experiment and collected the experimental data. X.Y., and Y.Z. conducted the biocompatibility test and discussed the experiments. C.L., S.Q., Z.J., and N.Z. analyzed the data and wrote the first draft. X.M. assisted in the additional revised data collection and analysis. C.L., S.Q., X.M., and N.Z. revised the manuscript. N.Z. supervised the project. All the authors discussed the results and commented on the manuscript.

## Competing interests

N.Z., C.L., Z.J., S.Q. and Y.Z. are co-inventors on a patent related to the technology (US Patent App. 63/710,715) described in this work. The other authors declare no competing interests.
