## [Transparent Peer Review file · Nature Communications]

A Conformal Piezoelectric Microsystem for Demographic-Adaptive and Calibration-Free Cuffless Blood Pressure Monitoring

Corresponding Author: Professor Ni Zhao

Version 1:

Reviewer comments:

Reviewer #1

(Remarks to the Author)
Comments to the Authors

Very nice article, and very important for the application of ultrasound and PWV for continuous and calibration-free blood pressure monitoring. However, I have some concerns about some parts of the article, that need to be reviewed and further explained:

1. In this article, you improve blood pressure monitoring by using a combination of two pulse waveform sensors and an array of 2x4 ultrasound sensors. Ultrasound sensors can be also used to determine the PWV, so why not use just the ultrasound sensors of the 2x4 array to calculate local PWV, so that you have only one device modality (ultrasound transducers to determine PWV and arterial distension at the same time)?

You can find an example of this approach in the following article: DOI: 10.1109/TBME.2024.3514878

Introduction

2. Line 40: Another important reference that is missed here about advanced BP models (applied in the field of ultrasound), and that also use PWV and diameter as input parameters, is this one:

doi: 10.1109/JSEN.2022.3215597

3. Line 109: Please include the definition of "Grade A" in your article, and check properly if your measurements can be categorized as "Grade A". If you check the following article:

<https://doi.org/10.1161/HYPERTENSIONAHA.117.10237>

You will notice that you need a sample size of $n=85$, and the study should include >30% males and >30 % females. Maybe publications do not take that into account, but if you are using smaller sample sizes is just wrong to say that the device is "Grade A".

Moreover, if you check in the Supplementary of this article (table S11):

doi: 10.1109/RBME.2024.3410399

You will notice that grade A is provided when the difference between the reference and test device is lower than 15 mmHg by 95% of the sample size. However, in your results from Supplementary Table 2, the SBP does not pass this condition (93.55 %). Please rewrite the sections where you mention "Grade A" properly.

Results

4. Equation (3): I read Supplementary Note 4 regarding the derivation of this equation, and also reference [4]. However, I still do not understand how you obtain $a=0.04$ and $b=0.04$ as demographic constants derived from clinical data. Did you do a linear fitting that relates AASI and PWV to determine a and b ? If so could you plot a graph with the data, or just explain how did you obtain those values more in detail?

Similarly, for a better understanding, you need also to introduce a definition of the variables $a_1, b_1, a_2, b_2, a_3, b_3, a_4, b_4$ in the Supplementary Table 2. Those variables are not defined anywhere.

5. Line 176: "while maintaining sufficient measurement accuracy". Please mention here the accuracy that you obtained (in value).

Device Design and Fabrication

6. Lines 193-194: self-powering capability. This is not mentioned anymore in the article, could you explain a bit more on that, and describe how the sensors have self-powering capability?

7. Lines 202-203: Strain pictures of the piezoelectric elements from Fig. 2c and Supplementary Fig. 3 are different, but the description in the figures is the same. What is the difference between those pictures? Are they measurements or simulations? It will help the reader to understand it properly if you better explain the caption of Supplementary Fig. 3

8. Line 212: The text says "higher resonant frequency compared to traditional PZT materials". As I understood from the text, your resonance frequency is 5 MHz, which is the typical frequency for PZT materials to measure blood pressure at the radial or carotid artery (some of the PZTs are working at a resonance frequency of 7 MHz, and even 10 MHz. Check here: DOI: 10.1126/science.abo2542). Please rewrite accordingly.

9. Line 214: "low acoustic impedance (15.3 MRayl) enables superior acoustic coupling with soft biological tissues compared to isotropic piezoelectric materials". The tissue has an acoustic impedance of circa 1.5 MRayl, so the coupling is not perfect. Please introduce the value of acoustic impedance for the tissue and for the isotropic piezoelectric materials. Are you including an impedance matching or coupling agent to fit the 15.3 MRayl from your PZTs to the tissue? Are the transducers focused or unfocused?

10. Line 238: "comfortable attached to human skin for long-term BP monitoring". How do you position and attach the sensor to the right position on the skin? How do you guarantee that the sensor is placed in the right position to measure, for example, the radial artery? This is usually the main challenge point in ultrasound, as the signal changes drastically depending on the position, so it is important to include some words about that in the article. Is it not manually positioned? (see line 249)

Device Characterization

11. Line 245 and Fig. 3a: "resonant frequency of 5 MHz". How much is the impedance at the resonance frequency? It is not easy to see the exact value in the figure. Moreover, it looks pretty high (around 200 Ohms), and your Pulser/Receiver should have an impedance of around 50 ohms, so you will lose a lot of signal through that connection. Are you applying an electrical impedance matching to reduce this effect?

12. Line 259: Crosstalk level around -57 dB. At which frequency did you measure that crosstalk? The crosstalk may change with the frequency.

Performance Validation

13. Line 299: the name and the company of the commercial ultrasound imager that you used here are missing.

14. Line 301: diameter measurements results being 2.5 mm and 2.53 mm. What is the resolution of each of the measurement methods? The reason for that deviation should not be due to the measurement at different times, but that you are using different devices, as they have different resolutions. The value that you give to the acoustic speed in tissue may also influence that, as it may be different from the acoustic speed of your reference device.

15. Line 309-310: "Posterior wall of the artery exhibits a wider range of motion compared to the anterior wall". I doubt that this is due to the lightweight nature of the device. In Supplementary Fig. 24, we see that the range motion difference is the lowest at the temporal artery and the highest at the brachial artery. The depth of those arteries is as follows: Temporal < Radial < Brachial. And about the diameter Temporal < Radial < Brachial. If you are always using the same frequency (5 MHz) with the ultrasound sensor, I guess that more superficial arteries as the temporal and radial arteries will be measured with more accuracy, than deeper ones (brachial). Also, with a bigger lumen diameter you will have more attenuation in the wave.

16. Line 330: "with a mean PWV of 5.11 m/s". Please include also the standard deviation.

17. Fig. 4e: Please include the temperature that you have on the white cursor for the thermography (on the bar between 17.2 and 31.4°C). Do this also as well in Suppl. Fig. 26 for the four pictures, so that we exactly know the temperature that you measured.

BP Monitoring

18. Line 400: "Supplementary Tables 2-3". Please include what the abbreviations m.e. and s.d. mean. What is the difference between m.e. and MAE? As I understood, you measured the waveforms as depicted in Supplementary Fig. 32 and 33, and for each measurement point, you determined the error to the reference device. From all those error points you obtained m.e. and s.d. Is this right?

19. Line 408: "The average m.e. and s.d. were -0.56 ± 3.36 mmHg for SBP". However, if we look at Suppl. Table 4, for Subject No. 12 the value of SBP is -10.10 mmHg, which does not fit within the mean value and the standard deviation mentioned above. I guess you just did the average of all the s.d. from all the subjects, which is not the right method to see if the calibration accuracy is within the standards. You need to plot the values of SBP and DBP (from your sensor and the reference device) in a Bland-Altman plot and determine the mean and the standard deviation out of this plot. This will give you the right accuracy value. You can follow as an example this article, or the article that you mention in Reference [23]: <https://doi.org/10.1002/adma.202301627>

I recommend doing this at least with Suppl. Table 4, and Suppl. Table 5. I think that will change your results in the text and also in Supplementary Table 8. Please look into this carefully.

Supplementary Table 2

20. What do you mean by "single and dual-modality" with 1-element model, 2 and 3-element model? Please just write which are the elements for each Blood Pressure Model in the caption.

Supplementary Table 8

21. Accuracy: The accuracy of Reference [18] is reported in the Supplementary of Reference [18]. Please include the values. Also for Reference [24] the standard deviation (4.92) is wrong. Please check the article and correct it.

22. In general, in the Supplementary Material, the equations do not have any number, please include the numbers of the equations, and define the variables and acronyms that are not defined (some definitions are lacking).

Reviewer #2

(Remarks to the Author)

This study presents a novel conformal and stretchable piezoelectric microsystem (CSPM) combined with a demographic-adaptive model for calibration-free cuffless blood pressure (BP) monitoring. This is a highly significant topic within this domain. This work addresses critical challenges in long-term BP tracking, such as frequent recalibration and signal drift, by synchronously measuring pulse wave velocity (PWV) and vascular diameter changes in the same arterial segment. While I have some concerns as following:

1. This study proposes synchronized PWV and vascular diameter measurements, I think the readers may wonder to clearly differentiate its approach from existing techniques (e.g., PTT or only PPG-based methods). So quantitative analysis may be provided to compare its parameter efficiency or computational complexity with state-of-the-art methods. And this may enhance the innovation and appeal of the article.

2. While the multilayer design is technically sophisticated, are there experiments tested on manufacturing reproducibility, batch consistency, and long-term durability under real-world conditions (e.g., sweat, motion artifacts, individual differences)? Mechanical stress tests (e.g., stretching up to 40%) are commendable but lack quantification of signal stability over repeated strain cycles. Some data may be needed.

3. If I understand in right way, the validation cohort (25 subjects, 5 excluded for calibration) is small for demographic generalization, Subgroup analyses (e.g., 7 females, 2 dark-skinned individuals) may lead to lack statistical power, raising concerns about algorithmic bias. Some discussion is needed. Long-term tracking (7 days) involves 2 subjects, how to define the robustness across diverse activities (exercise, sleep) or physiological states (e.g., hypertension)?

Also I have the mentioned concerns, this work do attract my attention. The CSPM represents advancement in cuffless BP monitoring, combining innovative hardware design with adaptive algorithms.

Reviewer #3

(Remarks to the Author)

This paper presents a holistic approach to address the challenging issue of time-varying physiological factors in continuous blood pressure (BP) monitoring.

(1) The current Results section is not sufficiently reader-friendly, as it includes content that belongs in the Methods section. For instance, the explanation of the BP principle should be placed exclusively in the Methods section. The authors should consider reorganizing the content to ensure that appropriate information is presented in the corresponding sections. Detailed elaborations, such as Equations (1) to (3), may hinder readers' ability to grasp the core findings. To improve clarity, the authors are encouraged to concisely summarize the underlying assumptions and fundamental principles at the outset, rather than presenting them in their current form. Overall, the authors should reconsider the structure and rearrange the content for better readability.

(2) I found it challenging to navigate between the manuscript and the supplementary material to fully understand the core ideas of BP estimation. Some parameters, such as blood linear density, are unconventional and require clarification. Before delving into detailed descriptions, the authors should summarize their underlying assumptions, clearly distinguish between measured and estimated parameters, and illustrate the relationships between them using visual aids such as flowcharts.

(3) A key point that needs further elaboration is why the linear relationship between pulse wave velocity (PWV) and PWV_CSPM is considered satisfactory. This should be explicitly justified to strengthen the manuscript's credibility.

(4) While the use of PWV measurements and vessel diameter is explained, the discussion would benefit from additional details on the precise calibration procedures. Providing more information on this aspect would enhance the methodological rigor of the study.

(5) The study relies on calibration data from a small cohort of participants ($n = 5$) to establish relationships between parameters such as PWV, β stiffness, and BP. The manuscript should include a more in-depth discussion on the implications of this limited sample size, particularly how it might impact the robustness and accuracy of the proposed method when applied to a larger and more diverse population.

Version 2:

Reviewer comments:

Reviewer #1

(Remarks to the Author)

Dear Authors,

thank you very much for addressing my comments. All the questions were in detail and properly addressed, increasing the impact of the article. The results are novel and very useful for the cardiovascular community, in terms of prevention of cardiovascular diseases via blood pressure measurements. I do not have any further comments. Nice job!

Reviewer #2

(Remarks to the Author)

The revised manuscript addresses most of the first-round review concerns (e.g., sample size expansion, manufacturing reproducibility), but some gaps remain in vascular depth adaptability, database transparency, manual positioning quantification, algorithm interpretability, and long-term wearability. These issues may be addressed to fully establish the CSPM's value. Further experiments (e.g., inter-operator positioning tests) and additional data disclosure (e.g., database details,) are recommended.

Response Letter

We sincerely appreciate the editor and reviewers for their valuable comments and thoughtful suggestions, which have significantly contributed to improving the quality of our manuscript. Based on these comments and suggestions, we have made careful modification and revisions. All changes are highlighted in the revised manuscript. Below are the point-by-point responses to the reviewers' comments.

Reviewer #1 (Remarks to the Author):

Comment:

Very nice article, and very important for the application of ultrasound and PWV for continuous and calibration-free blood pressure monitoring. However, I have some concerns about some parts of the article, that need to be reviewed and further explained:

[Authors' Response]: Thank you for your kind and encouraging feedback on our manuscript. In response to your concerns, we have carefully reviewed and addressed each of the points raised, as detailed below.

(1) In this article, you improve blood pressure monitoring by using a combination of two pulse waveform sensors and an array of 2x4 ultrasound sensors. Ultrasound sensors can be also used to determine the PWV, so why not use just the ultrasound sensors of the 2x4 array to calculate local PWV, so that you have only one device modality (ultrasound transducers to determine PWV and arterial distension at the same time)?

You can find an example of this approach in the following article: DOI: 10.1109/TBME.2024.3514878

[Authors' Response]: Thank you for your comments. While ultrasound sensors can indeed be used to calculate pulse wave velocity (PWV), they are not suitable for accurately extracting local PWV due to their limited temporal resolution. Below, we provide a detailed analysis illustrating why this approach is not feasible:

Ultrasound sensors typically operate by emitting and receiving ultrasound waves for measurements. In arterial diameter measurements, the common A-mode ultrasound measurement technique involves a pulse repetitive frequency (PRF) dictating the emission and reception of ultrasound waves. The PRF of these ultrasound systems, including the setup described in the referenced article (DOI: 10.1109/TBME.2024.3514878), is in the kilohertz range (e.g., 2 kHz). With a PRF of 2 kHz, the resulting time interval between measurements is **0.5 ms**, as shown in **Figure R1**. Local pulse transit time (PTT), which is essential for calculating local PWV, typically ranges from **2 to 5 ms** [1]. With a temporal resolution of 0.5 ms, current ultrasound systems cannot reliably capture the small time differentials required to accurately measure local PTT over short arterial distances.

Figure R1. Time resolution of pulse waveform obtained by ultrasound sensor.

Pressure-type pulse waveform (PWF) sensors, on the other hand, offer significantly higher temporal resolution. In our study, the PWF sensors operated at a sampling frequency of 315 kHz, corresponding to a sampling interval of approximately 3.2 μs (**Figure R2**). This is orders of magnitude higher than the temporal resolution provided by ultrasound sensors, enabling precise measurement of PTT and, consequently, local PWV.

To achieve the required accuracy in local PWV measurements, we opted for pressure-type PWF sensors instead of relying solely on ultrasound sensors. By analyzing the pulse waveforms from the two PWF sensors placed 12 mm apart, we calculated local PWV as the ratio of the distance to the time differential between the signals. This approach leverages the superior

temporal resolution of pressure sensors to ensure precise local PWV measurements, which would not be feasible using current ultrasound hardware.

Figure R2. Time resolution of pulse waveform obtained by the PWF sensor (pressure sensor) in our work.

Therefore, to clarify the consideration of PWV measurement by the designed sensing system, we have added the corresponding description to the revised manuscript (Please see Page 7, Lines 179-181).

“By analyzing these pulse waveforms, we can calculate the local PWV by dividing the 12 mm distance by the time differential between the signals from the two PWF sensors. We chose a pressure sensor as the PWF sensor because of its high temporal resolution.”

Reference

[1] Zhou J, Qu M, Liu W, et al. Continuous Monitoring of Blood Pressure by Measuring Local Pulse Wave Velocity Using Wearable Micromachined Ultrasonic Probes. *IEEE Transactions on Biomedical Engineering*, 2025. DOI: 10.1109/TBME.2024.3514878

(2) Introduction

Line 40: Another important reference that is missed here about advanced BP models (applied in the field of ultrasound), and that also use PWV and diameter as input parameters, is this one:
doi: 10.1109/JSEN.2022.3215597

[Authors' Response]: Thank you for your comments. We have revised the manuscript and included the reference you provided in the revised manuscript. (Please see Pages 2, Line 40, and Page 31, Line 838)

(3) Line 109: Please include the definition of “Grade A” in your article, and check properly if your measurements can be categorized as “Grade A”. If you check the following article:

<https://doi.org/10.1161/HYPERTENSIONAHA.117.10237>

You will notice that you need a sample size of $n=85$, and the study should include >30% males and >30 % females. Maybe publications do not take that into account, but if you are using smaller sample sizes is just wrong to say that the device is “Grade A”.

Moreover, if you check in the Supplementary of this article (table SI1):

doi: 10.1109/RBME.2024.3410399

You will notice that grade A is provided when the difference between the reference and test device is lower than 15 mmHg by 95% of the sample size. However, in your results from Supplementary Table 2, the SBP does not pass this condition (93.55 %). Please rewrite the sections where you mention “Grade A” properly.

[Authors' Response]: Thank you for your insightful comments and for pointing out the specific requirements for achieving "Grade A" categorization. We have carefully reviewed the references you provided and acknowledge that, given the limited sample size and associated criteria, our study does not fully meet the requirements for classification as “Grade A”. Therefore, we have revised the manuscript to remove any claims related to "Grade A" categorization.

In the meantime, **to further enhance the reliability of our findings, we have expanded the sample size to 45 subjects.** While this still does not meet the requirements for "Grade A" categorization, the **additional data continues to demonstrate the accuracy and robustness of our device and algorithm, particularly in calibration-free blood pressure measurements across diverse populations.** We recognize the importance of validating our technology on a larger population to fully meet the standards for "Grade A" categorization. To this end, we plan

to scale up the production of the sensor patch and conduct further testing with a larger and more diverse population in future studies.

The corresponding updated results can be found in the Supplementary Information (Please see SI-Pages 47, 48, 55 and 56).

(4) Results

Equation (3): I read Supplementary Note 4 regarding the derivation of this equation, and also reference [4]. However, I still do not understand how you obtain $a=0.04$ and $b=0.04$ as demographic constants derived from clinical data. Did you do a linear fitting that relates AASI and PWV to determine a and b ? If so could you plot a graph with the data, or just explain how did you obtain those values more in detail?

Similarly, for a better understanding, you need also to introduce a definition of the variables $a_1, b_1, a_2, b_2, a_3, b_3, a_4, b_4$ in the Supplementary Table 2. Those variables are not defined anywhere.

[Authors' Response]: We appreciate your careful review and the valuable comment.

Regarding the determination of the demographic constants $a=0.04$ and $b=0.04$, we employed linear regression analysis based on the data from the referenced literature [Ref. 1], which is Ref. 48 in the manuscript.

As illustrated in **Figure R3** from ref. 1, we utilized an online data extraction software (<https://automeris.io/wpd/>) to accurately obtain the original data points. Subsequently, through a linear regression fitting procedure, we obtained the relationship between the variables. After statistical computation, the coefficients a and b were determined. Noted that these coefficients serve as the initial parameters for the demographic model, and subsequent alignment of the cfPWV with our measured PWV through a linear mapping will facilitate the adaptation of the BP model for population-based applications (see Methods and Results).

Figure R3. Relation between the AASI and pulse wave velocity in 166 volunteers [Ref.1].

Following your suggestion, **we have added the variables $a_1, b_1, a_2, b_2, a_3, b_3, a_4, b_4$ to the definition of these variables as blood pressure calibration coefficients, see the corner notes under SI table2 in the revised Supplementary Information (SI-Page 53).**

“* a_1 and b_1 are the calibration coefficients of the PWV -BP method based on the experiments^[19]; a_2 and b_2 , in the PWV^2 -BP method, depend on the material properties and geometry of the artery and are to be determined from the experiments^[20]; a_3 is the ratio of blood density and beta stiffness and b_3 is a constant shift between the model-based BP and measured BP of the PWV^2 -Diam to BP method^[21]; a_4 and b_4 are the calibration coefficients that related to elastic modulus (at zero pressure) and blood density of the PWV^2 -A to BP method^[20].”

Reference

[1] Li Y, Wang J G, Dolan E, et al. Ambulatory arterial stiffness index derived from 24-hour ambulatory blood pressure monitoring[J]. Hypertension, 2006, 47(3): 359-364.

(5) Line 176: “while maintaining sufficient measurement accuracy”. Please mention here the accuracy that you obtained (in value).

[Authors’ Response]: Thank you for your comments.

We included the accuracy values as the comment. **The specific modifications are as follows (Please see Page 6, Lines 164-167).**

“As will be demonstrated in later sections, the CSPM and corresponding BP algorithm can record BP values across different individuals (varying in gender, skin tone, and age) without the need for repeated calibration, while maintaining sufficient measurement accuracy (Mean Absolute Error, MAE=mean(abs(BPref-BPest), SBP=5.22 mmHg, DBP=4.57 mmHg) (Fig. 1c).”

(6) Device Design and Fabrication

Lines 193-194: self-powering capability. This is not mentioned anymore in the article, could you explain a bit more on that, and describe how the sensors have self-powering capability?

[Authors' Response]: Thank you for your comments.

We incorporated a description of the self-powering capability as suggested. **The added description in revised manuscript (see Page 7, Lines 184-187) is as follows:**

“Piezoelectric materials convert mechanical energy, such as pressure, directly into electrical energy through the piezoelectric effect. This mechanism eliminates the need for an external power source, enabling piezoelectric sensors to function as autonomous, self-powered devices.”

(7) Lines 202-203: Strain pictures of the piezoelectric elements from Fig. 2c and Supplementary Fig. 3 are different, but the description in the figures is the same. What is the difference between those pictures? Are they measurements or simulations? It will help the reader to understand it properly if you better explain the caption of Supplementary Fig. 3

[Authors' Response]: Thank you for your comments.

Fig. 2c displays the strain distribution of the PWF sensors, while Supplementary Fig. 3 displays the deformation distribution used to calculate the strain (strain is defined as deformation divided by the original dimension that changed). Both figures are based on simulations. **We have updated the manuscript to clarify the analysis of stress and deformation distributions (Please see Page 7, Lines 196-199), and revised the caption of Supplementary Fig. 3.**

“Based on the finite element analysis, stress and deformation distributions of press mode and self-adhesion mode PWF sensors were examined, revealing that self-adhesion mode PWF sensors within a cavity exhibit greater deformation and strain compared to press mode sensors.”

(8) Line 212: The text says “higher resonant frequency compared to traditional PZT materials”. As I understood from the text, your resonance frequency is 5 MHz, which is the typical frequency for PZT materials to measure blood pressure at the radial or carotid artery (some of the PZTs are working at a resonance frequency of 7 MHz, and even 10 MHz. Check here: DOI: 10.1126/science.abo2542). Please rewrite accordingly.

[Authors’ Response]: Thank you for your comments. We apologize for the ambiguity in the original description. What we intended to convey is that the resonant frequency of 1-3 composite piezoelectric materials is higher than that of traditional PZT materials with the same size.

We have clarified this point in the revised manuscript (Please see Page 8, Lines 209-210), as shown below:

“... and a higher resonant frequency compared to traditional PZT materials with the same size, making it ideal for high-precision measurements (Supplementary Fig. 5).”

(9) Line 214: “low acoustic impedance (15.3 MRayl) enables superior acoustic coupling with soft biological tissues compared to isotropic piezoelectric materials”. The tissue has an acoustic impedance of circa 1.5 MRayl, so the coupling is not perfect. Please introduce the value of acoustic impedance for the tissue and for the isotropic piezoelectric materials. Are you including an impedance matching or coupling agent to fit the 15.3 MRayl from your PZTs to the tissue? Are the transducers focused or unfocused?

[Authors’ Response]: Thank you for your comments.

The revised manuscript now includes the specific acoustic impedance values as follows:

“Additionally, the low acoustic impedance (15.3 MRayl) of the 1-3 composite enables improved acoustic coupling with soft biological tissues (~1.5 MRayl) compared to isotropic piezoelectric ceramics materials (e.g., 33 MRayl for PZT).” (Please see Page 8, Lines 211-213)

Considering the specific application in this study, we did not incorporate a specific matching layer in the transducer design. This decision was made based on the following reasons: (1) The 1-3 composite provides significantly improved acoustic matching with the tissue compared to

isotropic PZT. (2) Adding a matching layer would significantly increase the device's thickness, potentially compromising its conformability. (3) The support material of the electrode is a 12.5 μ m-thick polyimide (acoustic impedance: 3.16 MRayl), and the encapsulation layer is a 32 μ m-thick silicon rubber (Ecoflex 00-30, acoustic impedance: 1.06 MRayl), as well as an adhesion layer (Silbione, acoustic impedance: 1.06 MRayl), all of which contribute to impedance matching. The ultrasound transducers are unfocused, which is suitable for the artery wall-tracking application.

(10) Line 238: “comfortable attached to human skin for long-term BP monitoring”. How do you position and attach the sensor to the right position on the skin? How do you guarantee that the sensor is placed in the right position to measure, for example, the radial artery? This is usually the main challenge point in ultrasound, as the signal changes drastically depending on the position, so it is important to include some words about that in the article. Is it not manually positioned? (see line 249)

[Authors' Response]: Thank you for your comments. The sensor is manually positioned on the skin. The procedure is as follows: Initially, the approximate location of the radial artery is identified by finger sensing the pulse on the wrist. The sensor array is then brought into contact with the identified location, and the program begins tracking ultrasound signals from the sensor pixels. The array is gently and slowly slid perpendicular to the artery. When a maximal shift between two ultrasound echo peaks (corresponding to the anterior and posterior walls of the artery) is observed, the sensor is correctly positioned above the artery. Once the correct position is identified, the whole sensor patch is gently pressed onto the skin to fix it in place. Thanks to the small sensor spacing (1.6 mm) and the 2 \times 4 array design, this process is straightforward and allows for quick and accurate positioning of the sensor over the artery.

The corresponding description has been added to the Methods section of the revised manuscript to describe the positioning process. (Please see Page 23, Lines 686-693)

“The sensor is manually positioned on the skin. The procedure is as follows: Initially, the approximate location of the radial artery is identified by finger sensing the pulse on the wrist. The

sensor array is then brought into contact with the identified location, and the program begins tracking ultrasound signals from the sensor pixels. The array is gently and slowly slid perpendicular to the artery. When a maximal shift between two ultrasound echo peaks (corresponding to the anterior and posterior walls of the artery) is observed, the sensor is correctly positioned above the artery. Once the correct position is identified, the whole sensor patch is gently pressed onto the skin to fix it in place.”

(11) Device Characterization

Line 245 and Fig. 3a: “resonant frequency of 5 MHz”. How much is the impedance at the resonance frequency? It is not easy to see the exact value in the figure. Moreover, it looks pretty high (around 200 Ohms), and your Pulsar/Receiver should have an impedance of around 50 ohms, so you will lose a lot of signals through that connection. Are you applying an electrical impedance matching to reduce this effect?

[Authors' Response]: Thank you for your comments.

The impedance of the transducer at the resonance frequency is 205 ohms. As you pointed out, our Pulsar/Receiver system operates at an impedance of around 50 ohms, which could result in signal loss through the connection. Currently, we have not implemented electrical impedance matching. However, the signal quality remains sufficient for the purposes of this study, as detailed below:

We evaluated the pulse-echo response of the ultrasound transducer on an agarose-based hydrogel mimicking human tissue, revealing a spatial pulse length of approximately 1.32 μs and a sensitivity of 31% within a -6 dB bandwidth. This level of ultrasound intensity was found to be sufficient for measuring vessel diameters, particularly in the wrist area where vessels are located at a maximum depth of around 5mm. This adequacy of ultrasound intensity was further confirmed by the echo signals obtained from vessel measurements, as shown in Fig. 4b.

For the ultrasound wall-tracking applications in this study, we have maintained sufficient ultrasound intensity without employing electrical impedance matching. This approach ensures that our system can effectively measure vessel diameters in the wrist region, which can meet the

requirements of our study. For future investigations involving deeper ultrasound detection or imaging studies, we acknowledge that implementing electrical impedance matching will be necessary to optimize performance and minimize signal loss.

(12) Line 259: Crosstalk level around -57 dB. At which frequency did you measure that crosstalk? The crosstalk may change with the frequency.

[Authors' Response]: Thank you for your comments.

Since our ultrasound sensor mainly works at the resonant frequency, we only tested the crosstalk at the resonant frequency of 5 MHz. We have provided the information (**Please see Page 9, Lines 255-257**) in the revised manuscript.

“Crosstalk phenomena among elements in the transducer array were examined at 5 MHz (see Methods and Supplementary Fig. 19), demonstrating stable crosstalk levels averaging around -57 dB,”

(13) Performance Validation

Line 299: the name and the company of the commercial ultrasound imager that you used here are missing.

[Authors' Response]: Thank you for your comment. The name and the company of the commercial ultrasound imager (Resona 7, Mindray) have been added in the revised manuscript (**Please see Page 11, Lines 299-301**).

“We also measured the diameter of the artery at the same location using a commercial ultrasound imager (Resona 7, Mindray), and the result is illustrated in Fig. 4c, ...”

(14) Line 301: diameter measurements results being 2.5 mm and 2.53 mm. What is the resolution of each of the measurement methods? The reason for that deviation should not be due to the measurement at different times, but that you are using different devices, as they have different resolutions. The value that you give to the acoustic speed in tissue may also influence that, as it may be different from the acoustic speed of your reference device.

[Authors' Response]: Thank you for your comments.

First, we would like to provide information regarding the resolution of the measurement methods. Our ultrasound sensor operates at a sampling frequency of 312.5 MHz, resulting in an effective resolution of 4.928 μm for tracing the arterial wall position. For the commercial ultrasound imager used in this work, the resolution is not explicitly stated in the device specifications. However, based on the pixel size and other information, we estimate the resolution to be approximately 68 μm . In both cases, the acoustic speed in tissue is assumed to be 1540 m/s for resolution calculations.

Secondly, we would like to elaborate on the comparison experiment. Our sensor arrays do not perform imaging but operate at a high sampling frequency ($\sim\text{kHz}$), enabling the capture of detailed radial pulse waveforms, as shown in Figure 4d. In contrast, the commercial ultrasound imaging device has a low imaging rate, capturing only limited frames per second, which makes it unsuitable for radial pulse waveform extraction. When comparing the measurement results, we used a single snapshot of the diameter from the commercial imager and compared it with the diameter measured by our sensor array at the half-maximum radial pulse intensity. As shown in Figure 4d, our sensor array tracks diameter variations between approximately 2.48 mm and 2.59 mm, and the commercial device's measurement falls within this range. That's why we originally proposed that the deviation (2.5 mm vs. 2.53 mm) likely arises from the different time frames at which the diameters were extracted. We also acknowledge that the resolution of the two devices may play a role in the differing measurement results. **We have updated the manuscript to clarify the resolution and provide further details about the comparison experiment. (Please see Page 11, Lines 303-313)**

“Our ultrasound sensor operates at a sampling frequency of 312.5 MHz, resulting in an effective resolution of 4.928 μm for tracing the arterial wall position. For the commercial ultrasound imager, the resolution is approximately 68 μm based on the pixel size. Therefore the resolution of the two devices may play a role in the differing measurement results. Moreover, our sensor arrays operate at a high sampling frequency ($\sim\text{kHz}$), enabling the capture of detailed radial pulse waveforms (Fig. 4d). In contrast, the commercial ultrasound imager has a low imaging rate,

which makes it unsuitable for radial pulse waveform extraction. When comparing the measurement results, we used a single snapshot of the diameter from the commercial imager and compared it with the diameter measured by our sensor array at the half-maximum radial pulse intensity. As shown in Fig. 4d, our sensor array tracks diameter variations between approximately 2.48 mm and 2.59 mm, and the commercial device's measurement falls within this range."

(15) Line 309-310: "Posterior wall of the artery exhibits a wider range of motion compared to the anterior wall". I doubt that this is due to the lightweight nature of the device. In Supplementary Fig. 24, we see that the range motion difference is the lowest at the temporal artery and the highest at the brachial artery. The depth of those arteries is as follows: Temporal < Radial < Brachial. And about the diameter Temporal < Radial < Brachial. If you are always using the same frequency (5 MHz) with the ultrasound sensor, I guess that more superficial arteries as the temporal and radial arteries will be measured with more accuracy, than deeper ones (brachial). Also, with a bigger lumen diameter you will have more attenuation in the wave.

[Authors' Response]: Thank you for your comments. Upon reanalyzing the experimental results, we agree with your observation that the depth of the arteries significantly influences the differences in motion ranges between the anterior and posterior walls of the artery. Additionally, we acknowledge that other factors, such as artery diameter and the distance between the artery and supporting bone, may also contribute to these observed discrepancies.

The corresponding content has been revised in the manuscript (Please see Page 11, Lines 319-322) to reflect these findings. The updated text is shown as follows:

"Notably, our measurements reveal that the posterior wall of the artery exhibits a wider range of motion compared to the anterior wall for the radial and brachial arteries, while both walls of the temporal artery move by approximately the same amount. This discrepancy can be attributed to variations in vessel depth and diameter at different measurement locations."

(16) Line 330: "with a mean PWV of 5.11 m/s". Please include also the standard deviation.

[Authors' Response]: Thank you for your suggestion. The standard deviation, which is 0.91 m/s,

has been added in the revised manuscript (Please see Page 12, Lines 341-342)

“... with a mean local PWV of 5.11 m/s and standard deviation of 0.91 m/s.”

(17) Fig. 4e: Please include the temperature that you have on the white cursor for the thermography (on the bar between 17.2 and 31.4°C). Do this also as well in Suppl. Fig. 26 for the four pictures, so that we exactly know the temperature that you measured.

[Authors' Response]: Thank you for your suggestion. We have added the temperature readings for Fig. 4e. Additionally, similar temperature values have been provided for the four images in Supplementary Fig. 28 (Figure R4). (Please see Page 28 in the revised manuscript and Page 35 in the revised Supplementary Information)

Figure R4. - Supplementary Fig. 28. The thermal distribution of the device at the outset, after 1 hour, 2 hours, and 3 hours of operation.

(18) BP Monitoring

Line 400: “Supplementary Tables 2-3”. Please include what the abbreviations m.e. and s.d. mean.

What is the difference between m.e. and MAE? As I understood, you measured the waveforms as depicted in Supplementary Fig. 32 and 33, and for each measurement point, you determined the

error to the reference device. From all those error points you obtained m.e. and s.d. Is this right?

[Authors' Response]: Thank you for your suggestion. m.e. is the absolute mean error, s.d. is the standard deviation of the error, and MAE is the Mean Absolute Error. m.e., s.d. and MAE are calculated by the following equations: $m.e. = \text{mean}(BP_{\text{ref}} - BP_{\text{est}})$, $s.d. = \text{std}(BP_{\text{ref}} - BP_{\text{est}})$ and $MAE = \text{mean}(\text{abs}(BP_{\text{ref}} - BP_{\text{est}}))$, respectively.

Based on the waveforms in Supplementary Figs 32 and 33 (Supplementary Figs 35 and 36 in the revised Supplementary Information), the error can be calculated as $\text{Error} = BP_{\text{ref}} - BP_{\text{est}}$, and m.e. and s.d. were obtained from the calculated error points, as you commented. **The relevant content has been modified in the revised manuscript (Please see Page 15, Lines 445-448).**

“First, we calculated the average mean error ($m.e. = \text{mean}(BP_{\text{ref}} - BP_{\text{est}})$) and standard deviation ($s.d. = \text{std}(BP_{\text{ref}} - BP_{\text{est}})$) of BP measurements with individual calibration for each subject, as presented in Supplementary Table 4.”

(19) Line 408: “The average m.e. and s.d. were -0.56 ± 3.36 mmHg for SBP”. However, if we look at Suppl. Table 4, for Subject No. 12 the value of SBP is -10.10 mmHg, which does not fit within the mean value and the standard deviation mentioned above. I guess you just did the average of all the s.d. from all the subjects, which is not the right method to see if the calibration accuracy is within the standards. You need to plot the values of SBP and DBP (from your sensor and the reference device) in a Bland-Altman plot and determine the mean and the standard deviation out of this plot. This will give you the right accuracy value. You can follow as an example this article, or the article that you mention in Reference [23]: <https://doi.org/10.1002/adma.202301627>
I recommend doing this at least with Suppl. Table 4, and Suppl. Table 5. I think that will change your results in the text and also in Supplementary Table 8. Please look into this carefully.

[Authors' Response]: Thank you for your valuable comments.

Following your suggestion and the reference article provided, **we have added Bland-Altman plots for the data, including Supplementary Table 4 and Supplementary Table 5.** Consequently, we calculated the mean and standard deviations according to the plots and updated the content in Supplementary Table 8 accordingly. Please note that during revision we have

increased the number of subjects, and our current test sample size is 45 individuals. We appreciate your guidance, which has led to a more comprehensive evaluation of our calibration accuracy.

The corresponding description was added to the revised manuscript (see Pages 15 and 16, Lines 445-452) and Supplementary Fig. 40 (Figure R5) and Supplementary Fig. 41 (Figure R6) in Supplementary Information (see SI-Pages 47 and 48).

“First, we calculated the average mean error (m.e.= $\text{mean}(\text{BP}_{\text{ref}}-\text{BP}_{\text{est}})$) and standard deviation (s.d.= $\text{std}(\text{BP}_{\text{ref}}-\text{BP}_{\text{est}})$) of BP measurements with individual calibration for each subject, as presented in Supplementary Table 4. From the Bland-Altman plots in Supplementary Fig. 40, The average m.e. and s.d. were obtained as 0.52 ± 3.38 mmHg for SBP and 0.88 ± 2.75 mmHg for DBP. For measurements without individual calibration, the results for all the subjects are presented in Supplementary Table 5. From the Bland-Altman plots in Supplementary Fig. 41, the average m.e. and s.d. were obtained as 0.90 ± 7.14 mmHg for SBP and 1.77 ± 5.51 mmHg for DBP.”

(a)

(b)

(c)

(d)

Figure R5 - Supplementary Fig. 40. Bland-Altman plots to validate the accuracy for SBP (a) and DBP (b) across diverse subpopulations with individual calibration. Correlation plots of the reference SBP (c) and DBP (d) and the corresponding BP estimations by our BP model across diverse subpopulations with individual calibration.

(a)

(b)

(c)

(d)

Figure R6 - Supplementary Fig. 41. Bland-Altman plots to validate the accuracy for SBP (a) and DBP (b) across diverse subpopulations without individual calibration. Correlation plots of the reference SBP (c) and DBP (d) and the corresponding BP estimations by our BP model across diverse subpopulations with individual calibration.

(20) Supplementary Table 2

What do you mean by “single and dual-modality” with 1-element model, 2 and 3-element model?

Please just write which are the elements for each Blood Pressure Model in the caption.

[Authors' Response]: Thank you for your comments.

In Supplementary Table 2, we specified the elements corresponding to each Blood Pressure Model, and part of the Supplementary Table 2 is as follows:

Modality	Methods	Elements	Reference
Single-modality with 1-element model	$P_d e^{\beta \left(\frac{D(t)}{D_d} - 1 \right)}$	D(t)	[18]
	$a_1 \cdot PWV + b_1$	PWV	[19]
	$a_2 \cdot PWV^2 + b_2$	PWV	[20]
Dual-modality with 2-element model	$\frac{PWV^2}{\left[\frac{A(A - A_0)}{A_0^2} \right]} \cdot a_4 + b_4$	PWV, A	[20]
	$\frac{PWV^2 D_d}{D} \cdot a_3 + b_3$	PWV, D	[21]
Dual-modality with 3-element model	$\frac{PWV^2}{A} \cdot \mu + \theta$	PWV, A, μ	This work

(21) Supplementary Table 8

Accuracy: The accuracy of Reference [18] is reported in the Supplementary of Reference [18]. Please include the values. Also for Reference [24] the standard deviation (4.92) is wrong. Please check the article and correct it.

[Authors' Response]: Thank you for your comments. **According to your suggestion, we have included/updated the corresponding values in Supplementary Table 8.**

(22) In general, in the Supplementary Material, the equations do not have any number, please include the numbers of the equations, and define the variables and acronyms that are not defined (some definitions are lacking).

[Authors' Response]: Thank you for your comments. We have now numbered each equation for easy reference. In addition, we also provided definitions for all variables and acronyms in the supplementary materials.

Reviewer #2 (Remarks to the Author):

Comment:

This study presents a novel conformal and stretchable piezoelectric microsystem (CSPM) combined with a demographic-adaptive model for calibration-free cuffless blood pressure (BP) monitoring. This is a highly significant topic within this domain. This work addresses critical challenges in long-term BP tracking, such as frequent recalibration and signal drift, by synchronously measuring pulse wave velocity (PWV) and vascular diameter changes in the same arterial segment. While i have some concerns as following:

[Authors' Response]: Thank you for your encouraging comments regarding our work. We greatly appreciate your recognition of the significance of this study. In response to your concerns, we have carefully revised the manuscript point by point based on the reviewers' comments. Detailed responses to each of your concerns are provided below.

(1) This study proposes synchronized PWV and vascular diameter measurements, i think the readers may wonder to clearly differentiate its approach from existing techniques (e.g., PTT or only PPG-based methods). So quantitative analysis may be provided to compare its parameter efficiency or computational complexity with state-of-the-art methods. And this may enhance the innovation and appeal of the article.

[Authors' Response]: Thank you for your valuable comments.

In response to your suggestion, **we have conducted a quantitative comparison of our approach with existing blood pressure measurement techniques**, including PTT and PPG-based methods (which are subsets of the PWV method). This comparison is detailed in the newly added **Supplementary Table 8 (Table R1)** and partially summarized in **Supplementary Table 2**. It evaluates various aspects, such as time complexity, spatial complexity, degree of computational complexity, and population accuracy.

The results highlight that while our method exhibits slightly higher time complexity compared to the traditional PTT and PPG-based analytical approaches, it significantly outperforms them in terms of calibration-free measurement capability and population accuracy.

These advantages arise from the complementary information provided by the synchronized PWV and vascular diameter measurements, which enable a more comprehensive characterization of the cardiovascular system's physiological properties. Furthermore, the hardware complexity of our method remains comparable to that of existing techniques, ensuring its practicality and feasibility for real-world applications.

The corresponding description was added to the revised manuscript (Please see Page 16, Lines 473-479).

“In addition, we conducted a quantitative comparison of our method with existing blood pressure measurement techniques in terms of time complexity, spatial complexity, degree of complexity, and population accuracy, and so on (Supplementary Table 8). The results demonstrate that while our method shows a slightly higher time complexity, it significantly outperforms other methods in terms of accuracy in population. In terms of space complexity, our method remains comparable to existing techniques, ensuring its practicality and feasibility.”

Table R1 - Supplementary Table 8. Computational complexity comparison of the proposed dual-modality method with other continuous BP measurement methods.

Methods	Time complexity	Space complexity	Comments	Degree of Complexity	Accuracy* in population	Calibration-free?
$a_1 \cdot PWV + b_1$	Training: O(1) Test: O(N)	Parameter storage: O(1) Output storage-N samples: O(N)	1 multiplication (a × PTT) and 1 addition (+b)	low	low	No
$a_2 \cdot PWV^2 + b_2$	Training: O(1) Test: O(N)	Parameter storage: O(1) Output storage-N samples: O(N)	2 multiplications + 1 addition	low	low	No
$P_d e^{\beta \left(\frac{D(t)}{D_d} - 1 \right)}$	Training: O(K×N) (K iteration number)	Parameter storage: O(1) Output storage-N samples: O(N)	1 division + 2 multiplication + 1 subtraction + 1	median	low	No

) Test: O(N)		exponential operation + 1 addition			
$\frac{PWV^2}{\left[\frac{A(A-A_0)}{A_0^2}\right]} \cdot a_4 + b_4$	Training: O(K × N) (K iteration number) Test: O(N)	Parameter storage: O(1) Output storage-N samples: O(N)	Multiplication: 3 times (with PWV ² , A × (A-A ₀), ×a) Division: 2 times Addition/subtraction: 1 time each	median	low	No
$\frac{PWV^2 D_d}{D} \cdot a_3 + b_3$	Training: O(K × N) (K iteration number) Test: O(N)	Parameter storage: O(1) Output storage-N samples: O(N)	2 multiplications + 1 division + 1 addition	median	low	No
$\frac{PWV^2}{A} \cdot \mu + \theta$	Training: O(K × N) (K iteration number) Test: O(N)	Parameter storage: O(1) Output storage-N samples: O(N)	4 multiplications + 3 divisions + 2 additions + 2 subtractions + 1 logarithmic number	median	high	Yes

* The detailed accuracy comparison is provided in Supplementary Table 2.

O(1): constant complexity

O(N): linear complexity

O(K×N): quadratic complexity

(2) While the multilayer design is technically sophisticated, are there experiments tested on manufacturing reproducibility, batch consistency, and long-term durability under real-world conditions (e.g., sweat, motion artifacts, Individual differences)? Mechanical stress tests (e.g., stretching up to 40%) are commendable but lack quantification of signal stability over repeated strain cycles. Some data may be needed.

[Authors' Response]: Thank you for your valuable comments. We fully acknowledge that manufacturing reproducibility, batch consistency, and long-term durability under real-world

conditions are essential for translating the multilayer design into practical applications. Below, we provide detailed responses to the additional experiments we conducted to address these aspects.

To address the concern regarding manufacturing reproducibility, we have conducted a comparison of CSPMs produced in three separate batches and performed BP tests on the same individual using these devices. The results, presented in **Supplementary Fig. 38 (Figure R7)**, demonstrate consistent performance across different batches, highlighting the reproducibility of our device design. Nevertheless, we acknowledge that the current manual steps involved in the multilayer manufacturing process pose challenges for large-scale production. The manufacturing process, however, is fundamentally based on standard semiconductor manufacturing methods and deterministic transfer technology. With the development and implementation of suitable transfer printing equipment, this process could be scaled up, and exploring this direction is part of our future work.

Figure R7 - Supplementary Fig. 38. BP results of three different CSPMs.

To evaluate the impact of sweat on the device's performance, we conducted blood pressure tests under both dry and sweaty conditions. The results, presented in **Supplementary Fig. 39 (Figure R8)**, demonstrate that the presence of sweat causes minimal performance degradation. We acknowledge that bodily motions can affect the wiring connections to the external ultrasound processing unit, leading to signal disruptions. To address this limitation, we are actively working

on miniaturizing the processing circuitry to achieve complete wearable integration.

Figure R8 - Supplementary Fig. 39. BP results of CSPM with and without sweat.

To evaluate long-term durability, we used the old CSPM devices to measure blood pressure in a group of 20 subjects approximately six months after their fabrication. These results were compared to those obtained on the initial 25-subject test group, who were measured using the same batch of devices. The mean error of BP across this 6 month time window is presented in **Supplementary Fig. 42 (Figure R9)**. The results confirm the device's reliability to measure blood pressure for 45 subjects over six months.

Figure R9 - Supplementary Fig. 42. Mean error of BP across diverse populations

Furthermore, we supplemented mechanical stress testing with data on signal stability during repeated strain cycles. Specifically, the device was subjected to 1000 stretching cycles (strain: 0–40%, frequency: 0.5 Hz), and its resonant frequency was measured at intervals of 100 cycles. The results, presented in **Supplementary Fig. 21 (Figure R10)**, show that the device’s resonant frequency remained unchanged after 1000 strain cycles, with an average resonant frequency of 5.21 MHz and a standard deviation of 76.67 kHz. These results demonstrate that the device maintains signal stability even after repeated exposure to 40% strain, confirming its mechanical durability for applications requiring cyclic deformation.

We hope these additional results comprehensively address your concerns and further validate the robustness and practicality of our multilayer design.

Figure R10 - Supplementary Fig. 21. Resonant frequency of ultrasound sensor under repeated strain cycles

The corresponding descriptions have been added to the revised manuscript (Please see Page 15, Lines 445-452; and Page 16, Lines 454-458).

“First, we calculated the average mean error (m.e.= mean(BPref-BPest)) and standard deviation (s.d.=std(BPref-BPest)) of BP measurements with individual calibration for each subject, as presented in Supplementary Table 4. From the Bland-Altman plots in Supplementary Fig. 40, The average m.e. and s.d. were obtained as 0.52 ± 3.38 mmHg for SBP and 0.88 ± 2.75 mmHg for DBP. For measurements without individual calibration, the results for all the subjects are presented in Supplementary Table 5. From the Bland-Altman plots in Supplementary Fig. 41,

the average m.e. and s.d. were obtained as 0.90 ± 7.14 mmHg for SBP and 1.77 ± 5.51 mmHg for DBP.”

“Additionally, we used the old CSPM devices to measure blood pressure in a group of 20 subjects approximately six months after their fabrication. These results were compared to those obtained on the initial 25-subject test group, who were measured using the same batch of devices. The mean error of BP across this 6 month time window is presented in Supplementary Fig. 42. The results confirm the device’s reliability to measure blood pressure for 45 subjects over six months.”

(3) If i am understand in right way, the validation cohort (25 subjects, 5 excluded for calibration) is small for demographic generalization, Subgroup analyses (e.g., 7 females, 2 dark-skinned individuals) may lead to lack statistical power, raising concerns about algorithmic bias. Some discussion is needed.

Long-term tracking (7 days) involves 2 subjects, how to define the robustness across diverse activities (exercise, sleep) or physiological states (e.g., hypertension)?

[Authors’ Response]: Thank you for your valuable comments.

We have expanded the test cohort to include 45 individuals, comprising 23 males and 17 females. While the larger sample size resulted in slightly higher MAE — increasing from 4.83 mmHg to 5.22 mmHg for SBP and from 4.51 mmHg to 4.57 mmHg for DBP — these values remain well within the acceptable threshold of 7 mmHg, confirming that the measurement accuracy is sufficiently high for both males and females.

Regarding different skin tones, theoretically, skin tones should not impact the accuracy of our device, as it relies on mechanical principles (pressure and ultrasound) rather than optical measurements. Our results show consistent SBP measurement errors across the three studied skin tones (**Figure R11**). Nevertheless, we acknowledge that further testing with a larger and more diverse cohort would strengthen these findings and will be explored in future work..

Finally, we fully acknowledge the limitations of our long-term tracking experiment, as it involved only two subjects. Currently, our data acquisition system is expensive and bulky, which

prevents its deployment in home settings and limits our ability to conduct comprehensive seven-day tracking across a larger and more diverse population. Expanding long-term tracking to encompass diverse activities (e.g., exercise, sleep) and physiological states (e.g., hypertension) is a key priority in our future work.

Figure R11 - Fig. 5 (d) mean absolute error (MAE) of measured BP across diverse subpopulations, encompassing gender, age, and skin tone, which verifies the effectiveness of the proposed BP model.

We have incorporated these updates in the manuscript (Please see Page 15, Lines 426-438), and the revised content is as follows:

“Figure 5d shows CSPM performance across gender, age, and skin tone subpopulations. Similar blood pressure errors were observed in males (MAE: SBP = 5.34 mmHg, DBP = 4.55 mmHg, n=23) and females (MAE: SBP = 5.07 mmHg, DBP = 4.59 mmHg, n=17). These subtle discrepancies in error may stem from intrinsic variations in vascular parameters between the sexes, yet remain accurately measured by our CSPM. Similarly, younger participants (<26 years) exhibited slightly larger errors (MAE: SBP = 5.33 mmHg, DBP = 4.63 mmHg, n=12) compared to older participants (\geq 26 years; MAE: SBP = 5.01 mmHg, DBP = 4.41 mmHg, n=28). Across skin tones, consistent low BP errors were observed: light (MAE: SBP = 5.36 mmHg, DBP = 2.68 mmHg, n=4), dark (MAE: SBP = 5.43 mmHg, DBP = 3.32 mmHg, n=7), and medium (MAE: SBP = 5.19 mmHg, DBP = 5.06 mmHg, n=29). Theoretically, skin tones should not impact test results as our device operates on mechanical principles (pressure and ultrasound) rather than optical measurements. The results show similar SBP measurement errors across the three skin tones, while some discrepancies exist in DBP measurements.”

(4) Also i have the mentioned concerns, this work do attract my attention. The CSPM represents advancement in cuffless BP monitoring, combining innovative hardware design with adaptive algorithms.

[**Authors' Response**]: Thank you again for your support! We appreciate your interest in our work, and hope the revised manuscript properly address your concerns.

Reviewer #3 (Remarks to the Author):

Comment:

This paper presents a holistic approach to address the challenging issue of time-varying physiological factors in continuous blood pressure (BP) monitoring.

[Authors' Response]: Thank you for your supportive comments on our manuscript. We have carefully revised the manuscript point by point in response to your feedback, as follows:

(1) The current Results section is not sufficiently reader-friendly, as it includes content that belongs in the Methods section. For instance, the explanation of the BP principle should be placed exclusively in the Methods section. The authors should consider reorganizing the content to ensure that appropriate information is presented in the corresponding sections. Detailed elaborations, such as Equations (1) to (3), may hinder readers' ability to grasp the core findings. To improve clarity, the authors are encouraged to concisely summarize the underlying assumptions and fundamental principles at the outset, rather than presenting them in their current form. Overall, the authors should reconsider the structure and rearrange the content for better readability.

[Authors' Response]: Thank you for your valuable comments. We agree that certain content, particularly the detailed explanation of the blood pressure (BP) calculation principles, is more appropriate for the Methods section to enhance the readability of the Results section. **To address this concern, we have reorganized the manuscript and revised the relevant paragraphs, moving the detailed description of the BP calculation principles to the Methods section.** Additionally, **we have concisely summarized the underlying assumptions and fundamental principles at the beginning of the Results section** to provide readers with a clear understanding of the core findings

The specific modifications are shown as follows (Please see Page 5, Lines 109-137; and Page 19, Lines 540-558):

“Results

BP Measurement Principle

For BP measurement, according to the theory of elastic mechanics, the exact BP value of a section of

blood vessel can be calculated from the deformation and stiffness of the section of blood vessel. To satisfy the above theory, the following assumptions are made: (1) the blood vessel within the local area is uniform and its cross-section is circular; (2) the mechanical parameters of the blood vessel in the local area remain substantially unchanged (as illustrated in Fig. 1a).

In the following, we describe the underlying physical principles for calculating BP from local PWV and vascular diameter. BP refers to the force exerted by blood as it flows through blood vessels. Elastic arteries with high compliance can store greater volumes of blood, reducing the pressure impact on surrounding tissues, which in turn decreases PWV and subsequently lowers BP. Conversely, increased vascular stiffness reduces the pressure gradient and raises PWV, which may lead to elevated BP²⁴. Therefore, BP can be characterized by quantifying the volume of blood passing through a specific vascular segment under stiffness conditions (see **Methods** and **Supplementary Note 4** for details), expressed as^{44,45}:

$$BP = \frac{PWV^2 \cdot \rho}{A \cdot \beta} + \theta = \frac{PWV^2 \cdot \mu}{A} + \theta \quad (1)$$

where β is the beta stiffness related to the slope of pressure, A is vascular cross-sectional area, ρ is the blood density, $\mu = \rho / \beta$ represents the blood linear density defined as the rate of change in blood density along a unit length, and θ is linked to the intrinsic properties of the artery that maintain patency at low pressures.

From equation (1), accurately measuring BP requires obtaining physical parameters that represent vascular stiffness (PWV , β or μ) and deformation (cross-sectional area A). These parameters must be measured within the same local area to maintain the homogeneity of the vessel's material properties (e.g., Young's modulus and Poisson's ratio) and meet the uniformity assumption of the Bramwell-Hill equation⁴⁶. Furthermore, the radial muscular artery is the preferred site for observing dynamic blood volume distribution within the vascular system⁴⁷, particularly when assessing linear blood density, as discussed later in this study.”

“Methods

Details of BP calculation principle

While PWV and A can be measured directly, other parameters in equation (1) are obtained indirectly. The stiffness parameter β in equation (1) is an intrinsic characteristic of the blood vessel wall that indicates how vascular stiffness adjusts relative to changes in BP and cross-sectional area. Clinical observations have shown that pulsatile vascular stiffness varies with BP levels and correlates demographically with PWV ⁴⁸. Consequently, once PWV and A are determined by the CSPM sensors,

β can be estimated using the following equations:

$$\beta = \frac{\ln(PSR)}{\Delta A} \quad (2)$$

$$PSR = \frac{1}{1 - (a \cdot cfPWV + b)} \quad (3)$$

where ΔA is the maximum change in cross-sectional area during one cardiac cycle, PSR is the pulse stiffening ratio⁴⁹, and $a = 0.04$ and $b = 0.04$ are demographic constants that can be derived from clinical data⁴⁸ (see **Supplementary Note 4** for further details). $cfPWV$ is the carotid-femoral pulse wave velocity. Note that we aligned the $cfPWV$ with our measured PWV through a linear mapping of β -based μ (μ_β) to a reference μ derived by equation (1), details in results. Additionally, since blood density ρ shows considerably less variation than β stiffness in the general population (see **Supplementary Table 1**), the μ_β can be calculated given a known β under a fixed ρ , with ρ obtained directly from the literature⁵⁰.”

(2) I found it challenging to navigate between the manuscript and the supplementary material to fully understand the core ideas of BP estimation. Some parameters, such as blood linear density, are unconventional and require clarification. Before delving into detailed descriptions, the authors should summarize their underlying assumptions, clearly distinguish between measured and estimated parameters, and illustrate the relationships between them using visual aids such as flowcharts.

[Authors' Response]: Thank you for your comments.

Blood linear density is defined as the rate of change in blood density along a unit length. Elevated values indicate a greater blood density per unit length, signifying increased blood flow and, consequently, heightened lateral pressure on the vessel walls, resulting in elevated blood pressure.

As mentioned in our response to the previous question, we have revised the manuscript to include a concise summary of the underlying assumptions and fundamental principles at the beginning of the Results section (see **Page 5**). This provides readers with a clear understanding of the theoretical foundation before delving into detailed findings.

In addition, based on your suggestion, we have created a **flowchart** illustrating the calibration-free blood pressure calculation process. This flowchart, now included as

Supplementary Fig. 1 (Figure R12), visually represents the relationships between various measured and estimated parameters, such as pulse wave velocity (PWV), vascular diameter, and the derived blood pressure values.

The specific modifications are as follows (Please see Pages 5, Lines 111-115 in the manuscript; Pages 6, Lines 146-147; and SI-Fig. 1, Page 8 in Supplementary Information):

“For BP measurement, according to the theory of elastic mechanics, the exact BP value of a section of blood vessel can be calculated from the deformation and stiffness of the section of blood vessel. To satisfy the above theory, the following assumptions are made: (1) the blood vessel within the local area is uniform and its cross-section is circular; (2) the mechanical parameters of the blood vessel in the local area remain substantially unchanged (as illustrated in Fig. 1a).”

“Building on the above theory, we developed a workflow for calibration-free blood pressure calculation, as illustrated in the flowchart presented in Supplementary Fig. 1.”

Figure R12 - Supplementary Fig. 1. Flowchart of calibration-free BP calculation.

(3) A key point that needs further elaboration is why the linear relationship between pulse wave velocity (PWV) and PWV_CSPM is considered satisfactory. This should be explicitly justified to strengthen the manuscript's credibility.

[Authors' Response]: Thank you for your comment.

The linear relationship between pulse wave velocity (PWV) (measured using the gold standard, carotid-femoral PWV, cfPWV) and PWV_CSPM is considered a key foundation for our BP calculation principle, where PWV_CSPM is treated as the local PWV along the artery. This assumption is grounded in fundamental physiological principles, which suggest that a linear relationship exists between PWVs measured at different arterial segments. This linearity arises because arterial stiffness, a key determinant of PWV, is influenced by common physiological factors, such as vessel wall elasticity and composition, shared across arterial segments. [ref 1]. Previous studies have also demonstrated that different types of PWV can be linearly related [refs 2-3]. In our study, we observed a similar linear relationship between cfPWV and PWV_CSPM, as illustrated in **Supplementary Fig. 30 (Figure R13)**. Given the anatomical location of the CSPM system, we infer that PWV_CSPM can reliably represent the local PWV.

Furthermore, our follow-up experimental results validate this assumption. For example, we observed that μ_{PWV} (i.e., μ_{β} in the manuscript) can be linearly mapped to μ_{ref} , further strengthening the credibility of this relationship. These findings collectively justify the linear relationship between PWV and PWV_CSPM as satisfactory and foundational to our proposed method.

Figure R13 - Supplementary Fig. 30. Relationship between PWV and PWV_CSPM.

The corresponding description was added to the revised manuscript (Please see Page 13, Lines 356-359).

“Using commercial PWV test equipment and CSPM, we obtained a linear relationship between pulse wave velocity (PWV) and PWV_CSPM (**Supplementary Figs. 30**): $cfPWV = k \cdot PWV_{CSPM} + c$. Given the anatomical location of the CSPM system, we infer that PWV_CSPM can reliably represent the local PWV.”

Reference

- [1] Nichols, W. W., O'Rourke, M. F., & Vlachopoulos, C. (2011). McDonald's Blood Flow in Arteries: Theoretical, Experimental and Clinical Principles (6th ed.). CRC Press.
- [2] Chow B, Rabkin S W. Brachial - ankle pulse wave velocity is the only index of arterial stiffness that correlates with a mitral valve indices of diastolic dysfunction, but no index correlates with left atrial size[J]. Cardiology Research and Practice, 2013, 2013(1): 986847.
- [3] Sugawara J, Hayashi K, Yokoi T, Cortez-Cooper MY, DeVan AE, Anton MA, Tanaka H. Brachial-ankle pulse wave velocity: an index of central arterial stiffness? J Hum Hypertens. 2005 May;19(5):401-6. doi: 10.1038/sj.jhh.1001838. PMID: 15729378.

(4) While the use of PWV measurements and vessel diameter is explained, the discussion would benefit from additional details on the precise calibration procedures. Providing more information on this aspect would enhance the methodological rigor of the study.

[Authors' Response]: Thank you for your comments.

According to your suggestions, **we have added additional details and discussion on the calibration procedures in the revised manuscript (Please see Page 13, Lines 363-375).**

“Before performing the BP algorithm, the basic demographic characteristics are determined during initial calibration. Supplementary Figs. 31 and 32 show the five subjects of personalized blood vessel cross-sectional area A , PWV, β stiffness, and μ linear blood density, where β stiffness is calculated by Supplementary equation (6) and μ is calculated by substituting known

blood pressure, PWV, and A in equation (1). The subject-specific linear blood density was estimated directly by substituting the PWV and diameters in Supplementary equation (10), aligning well ($p < 0.0001$) with the reference linear blood density by substituting known blood pressure in equation (1). The calibration methodology presented herein focuses on the population parameter calibration, derived from and subsequently recalibrated using the dataset. This approach ensures that all model coefficients are determined from population-level information, thereby eliminating correlation with individual-specific data. This strategy offers the advantage of broad applicability across diverse populations, negating the need for subsequent recalibration. Specifically, for the coefficients outlined in equation (1), we employ population data for all calibration determinations ($n=5$).”

(5) The study relies on calibration data from a small cohort of participants ($n = 5$) to establish relationships between parameters such as PWV, β stiffness, and BP.

The manuscript should include a more in-depth discussion on the implications of this limited sample size,

particularly how it might impact the robustness and accuracy of the proposed method when applied to a larger and more diverse population.

[Authors' Response]: Thank you for your comments.

Our research currently relies on calibration data from a relatively small cohort of five participants to establish relationships between parameters such as pulse wave velocity (PWV), β stiffness, and blood pressure (BP). This decision was made because our original subject group consisted of only 25 participants, and we aimed to allocate a sufficient number of subjects for testing the accuracy of BP measurements. We are encouraged by the fact that, even with this small cohort, our method demonstrated promising BP measurement results.

As the reviewer pointed out, the limited sample size does present potential challenges to the generalizability and robustness of our findings. We acknowledge that a larger and more diverse cohort for initialization—particularly for deriving the demographic-adaptive constants used in BP calculations—would likely enhance the reliability and accuracy of our system. Expanding the

cohort remains a critical direction for future work, and we are confident that this step will significantly improve the performance and applicability of our method across broader populations.

In the revised manuscript (Please see Pages 17 and 18, Lines 517-536), we have added a more comprehensive discussion section to address these concerns in detail.

“Our study currently relies on calibration data from a relatively small cohort of five participants to establish relationships between parameters such as PWV, β stiffness, and BP. The limited sample size poses potential limitations to the generalizability of our findings. The current study’s 5-sample calibration size is suitable for small populations (dozens to hundreds of individuals) but insufficient for larger, more diverse ones. A small sample may miss extensive physiological variations in larger groups, harming the algorithm’s robustness, reducing prediction accuracy, and causing overfitting—making established correlations perform poorly on new individuals with different characteristics. Thus, calibrating sample size to the test population scale is critical, as a more appropriate size boosts the algorithm’s ability to handle variability and ensures reliable, accurate results.

Since small samples may not represent larger, diverse populations, calibration sample selection requires care. Instead of random sampling, samples should be deliberately chosen to cover a wide range of parameters (e.g., PWV, β stiffness, BP). This targeted approach helps the calibration process account for full physiological differences, improving the method’s generalizability, capturing complex parameter relationships, and enhancing the model’s adaptability to real-world scenarios.

For sufficiently large populations, using a small fraction of individuals for parameter calibration is an emerging trend in future large-cohort blood pressure measurement. Subsequent research will focus on exploring how to conduct model calibration with the minimum number of individuals while maintaining algorithm stability, involving the development of innovative sampling strategies, optimization techniques, and validation methods to balance sample efficiency and algorithm performance.”